# Family Tree for Aqueous Organic Redox Couples for Redox Flow Battery Electrolytes: A Conceptual Review

**DOI:** 10.3390/molecules27020560

**Published:** 2022-01-16

**Authors:** Peter Fischer, Petr Mazúr, Joanna Krakowiak

**Affiliations:** 1Fraunhofer Institute for Chemical Technology, Pfinztal, Joseph-von-Fraunhofer Str. 7, 76327 Pfinztal, Germany; 2Department of Chemical Engineering, University of Chemistry and Technology Prague, Technická 5, Praha 6, 166 28 Prague, Czech Republic; mazurp@vscht.cz; 3Physical Chemistry Department, Faculty of Chemistry, Gdańsk University of Technology, Narutowicza 11/12, 80-233 Gdańsk, Poland; joakrako@pg.edu.pl

**Keywords:** energy storage, flow battery, aqueous, organic redox compounds, electrolyte, energy density, power density, lifetime, cross-over, degradation

## Abstract

Redox flow batteries (RFBs) are an increasingly attractive option for renewable energy storage, thus providing flexibility for the supply of electrical energy. In recent years, research in this type of battery storage has been shifted from metal-ion based electrolytes to soluble organic redox-active compounds. Aqueous-based organic electrolytes are considered as more promising electrolytes to achieve “green”, safe, and low-cost energy storage. Many organic compounds and their derivatives have recently been intensively examined for application to redox flow batteries. This work presents an up-to-date overview of the redox organic compound groups tested for application in aqueous RFB. In the initial part, the most relevant requirements for technical electrolytes are described and discussed. The importance of supporting electrolytes selection, the limits for the aqueous system, and potential synthetic strategies for redox molecules are highlighted. The different organic redox couples described in the literature are grouped in a “family tree” for organic redox couples. This article is designed to be an introduction to the field of organic redox flow batteries and aims to provide an overview of current achievements as well as helping synthetic chemists to understand the basic concepts of the technical requirements for next-generation energy storage materials.

## 1. Introduction

During the last 20 years, renewable energies have been a key element for the subsequent transition of ignition-based power plants to zero carbon emission power sources. Due to the intermittency of the renewable energies, fast and reliable energy storage devices are required for the development of the renewable energy power supply with a longer service life without putting pressure on the Earth’s resources [1,2]. The revolutionary aspect of this transition is the opportunity for fundamental change in the electrical energy supply: the transition from a centralized controlled electricity supply to decentralized energy production. The localized production of energy comes at a certain cost, as the control requires flexibility in generation to meet the demand. It is widely accepted that battery storage is the easiest and the most efficient solution to match supply and demand and therefore is regarded to play a vital role in this transition. Nevertheless, long-duration battery storage is an expensive technology and usually comes with limitations in lifetime. This is one reason why grid-connected battery installations with cycle times of several hours are still not that widespread as the forecasted storage demand needs to be fulfilled. To understand the pivotal gap of installed storage units, the economy of battery storage has to be considered. Today, the cost of daily energy demands certainly depends on the tariff structure in each country, but only in seldom cases do the revenues for grid-connected batteries with long duty cycles match the costs of storage over the whole battery’s life expectancy. Additionally, batteries with intercalation electrodes (such as lithium ion batteries) or electrodes with low-conducting reaction products’ (e.g., lead sulfate) deep cycling operation reduces their life expectancy significantly. This is the reason why most battery storage solutions installed today address the more lucrative power market, which offer higher revenues and, due to very short charge and discharge cycles in a narrow range of SoC, and also offer the advantage of a prolonged battery lifetime.

All-vanadium redox flow batteries (VRFB) are regarded as one of the most commercialized mid- to long-duration battery storage solutions. The biggest long-duration projects in China have been built lately with VRFB technology [1]. However, the limiting costs of VRFB depend strongly on the electrolyte price. The vanadium electrolyte provides a lower threshold of the CAPEX cost of VRFB of 60–80 $/kWh, calculated with a mere raw material price of around 6–8 $/Ib V_2_O_5_. With current raw material prices, a VRFB cannot be built below this price. There are also several inorganic alternatives to VRFB with potentially significantly lower capacity costs, such as hybrid flow batteries based on iron [1] and zinc [3]. However, the increased complexity of this storage due to metal deposition and/or the involvement of gas phase active species (in metal-air systems) decreases the efficiency and operating lifetime of current systems significantly when compared to VRFB. Organic electrolytes could change the cost structure of long-duration RFB storage, as they offer the potential for cheap raw material and further cost reduction due to large-scale synthesis [4]. In recent years, commercialization efforts have increased, since start-ups, such as JenaBatteries (GE), Kemiwatt (FR), XL Batteries (USA), CM-Blu (GE), and Green Energy Storage (IT), appeared on the market. These start-ups are founded by university research groups, who had their own development in aqueous organic redox couples, such as TEMPO-Viologen at the Friedrich-Schiller University of Jena or quinone-type batteries at Harvard University and the University of Rennes. Currently, the benefits from these potentially cheaper redox chemistries have not displaced VRFB as the major flow battery chemistry in the market, as there are still some hurdles to solve before gaining a wider market share. However, as these electrolytes develop further, they will become increasingly more competitive in the future. The biggest problem facing aqueous organic electrolytes so far is the comparably low energy density, as mixed electrolytes of aqueous organic redox couples hardly exceed 13 Wh/L (≈1 mol/L e^−^) [5]. In contrast, the state-of-art inorganic RFB systems provide significantly higher values, such as 25–35 Wh/L for VRFB and around 70 Wh/L for the Zn-Br system [6], due to the typically higher concentration of active species and nominal battery voltage. Within this review, the term “mixed electrolyte” is used for a solution containing both electroactive species for the positive and negative electrode. Unmixed organic RFB electrolytes with higher energy density (≈20–40 Wh/L), on the other hand, suffer from a time-dependent capacity decay over time due to active species cross-over, as discussed in detail in Section 2.8. The perfect selective membrane with low ohmic resistance has not been developed so far, and as a result, the battery chemistry mixes over time, leading to an observable capacity decay.

The second obstacle facing aqueous organic electrolytes is the chemical instability of the applied redox molecules. Once the redox moiety has been destroyed, it is nearly impossible to recover the capacity. As a result, new battery chemistries need further investigations to resolve its actual lifetime and operational costs. Thus, the question of whether aqueous organic electrolytes may replace vanadium electrolytes still remains open. So far, all attempts to consider a non-aqueous flow battery chemistry based on organic solvents, deep eutectics, or ionic-liquid electrolytes have failed in industrial application due to the relatively low power densities at room temperature (usually not more than 1–2 mW/cm^2^, see Section 2.1), high price, and safety issues originating from the non-aqueous phase. This is why in this article the authors focus solely on aqueous flow battery electrolytes.

The aim of this work is to summarize promising redox motives for synthetic chemists and to review some important aspects of the development of technical redox flow battery electrolytes. The requirements are presented and potential synthetic strategy lines for designing new redox active components are shown. Additionally, the current limits of the aqueous organic electrolytes are briefly discussed and some general recommendations for future R&D directions are formulated.

### Redox Flow Batteries

In order to understand the functions of a battery in general, it has to be viewed as an electrochemical reactor. From this perspective, the battery poles are merely connectors to the reaction surfaces, called electrodes. The magic of energy storage happens on these two interfaces, which form a reaction surface for the plus and the minus pole battery reactions. The two electrodes are usually dissected into two half-cells by a separator, which is electrically insulating, but acts as an ionic conductor to balance out the charges converted in the electrochemical reactions. In some cases, this is just a porous insulating material while in other cases, the separator needs to have the ability to select certain ions by charge or size. If redox-active substances are present at the electrodes, a potential difference builds up, which is called the open circuit voltage (OCV). As soon as the poles are connected to a voltage drain or source, an electric current can flow in the form of electrons via the external circuit and in an ionic and electronic form within the cell via an electron transfer reaction (called redox reaction), converting chemical into electrical energy or vice versa. During the conversion, the charges exchanged between the two reaction surfaces (electrodes) lead to changes in the redox state of the storage material. The circuit is closed by an ionic current in the liquid junction of ion-rich solutions, which is typically referred to as electrolyte. To allow a continuous reaction, the transport of charges via the electrolyte has to be optimized to achieve low internal resistances in the cell. The equivalence of flown charges and converted number of molecules is expressed by adaptations of Faradays law of electrolysis (1). In a regular “static” battery cell, the electrolyte has the sole role of transferring ionic charges to balance the charges produced on the electrodes during the electrochemical conversion and, by this, closing the electrical circuit. In the discharge case, which is usually the value-generating part of battery operation, the electrode where oxidation of the reduced form of the negative active species proceeds is referred to as the anode (or more accurately the negative electrode), while the electrode where the reduction of the oxidized state of the positive active species proceeds is referred to as the cathode (or the positive electrode).

A regular “static” battery cell can be seen as an electrochemical batch reactor in this regard, as the reaction medium is confined in a defined space localized within the battery cell. A redox flow battery, on the other hand, can be seen as a continuous reactor [7]. Two electrolytes, one for the anodic and one for the cathodic reaction, contain the stored energy in a fluidized form. Analogically, the negative electrolyte is called negolyte (or sometimes anolyte) and the positive one posilyte (or sometimes catholyte). They are stored in two tanks and are pumped through the reactor and converted there, making the electrolyte not only a mere charge carrier, but also a storage medium. This additional feature of the electrolyte has some implications in regards to the solubility of active storage medium and conducting salts, which will be explained later in the text.

A technical electrolyte has to ensure stable battery operation in a given temperature window (usually 5–40 °C or broader). Technical electrolytes also have to fulfill some performance criteria to be used in an industrial setting. As a rule of thumb, minimum targets for electrolytes should (a) allow power densities above 50 mW/cm^2^ in a cell, (b) solubilities of redox active material above 1 mol/L electron equivalents, (c) dynamic viscosities below 10 mPa∙s, (d) cell voltages of more than 1 V, and (e) lifetime above 6000 full SoC range cycles at rated power. The last target should of course include the possibility of regeneration cycles to regain capacity, which is a common practice in technical flow batteries.

These are roughly the minimum requirements for considering these electrolytes in a technical application. These values may appear a bit arbitrary and, for sure, there might be some technical aqueous flow battery chemistries that have lower values in one of these categories. However, has to be noted that flow battery chemistries with lower value propositions than this might need convincing arguments regarding why they should be considered for a technical scale-up. For example, it is highly unlikely that non-aqueous batteries will meet all these criteria in the near future, as, e.g., the conductivity of nearly all known electrolytes of organic solvents are already several orders of magnitude apart from aqueous electrolytes, and thus the resulting power density will be too low to be considered in a technical scale-up.

In principle, all soluble ions or molecules with more than two oxidation states can be used as electrolyte. However, the combined application as the storage medium and ionic junction in a battery cell requires some chemical and physical properties of the redox media. In this review, we define 10 criteria for technical electrolytes that have an effect on the most important key performance indicators of the battery system, which are the (a) energy density, (b) power density, (c) lifetime, (d) environmental impact, (e) efficiency, and (f) price.

In the next paragraphs, the requirements for technical flow battery electrolytes are summarized and examples are provided, which we aim to link to these 10 criteria provided in Table 1.

## 2. Requirements for Organic Redox Flow Battery Electrolytes

The electrolyte in a flow battery has to fulfill several tasks. It is (a) an energy storage medium, (b) charge reservoir for charge exchange via the separator, (c) ionic conductor, and (d) fluidic transport medium. With this comes several parameters (see Table 1), in which the electrolyte has to be optimized.

### 2.1. Solubility

Aqueous organic electrolytes are based on organic redox couples dissolved in water and a supporting electrolyte. Water is one of the few solvents that allows solubility of high concentrations of charge ions due to its unique solvation properties, e.g., the water molecule has one of the highest permeability (ε), the highest acceptor numbers (ANs), as well as the highest polarizability (π*) of all known solvents. Water has also a high donor number and moderate viscosity as shown in Table 2. The other advantage is that water is a small molecule, leading to relatively small solvated ions. This property is beneficial for the viscosity and other rheological parameters of highly concentrated solutions.

The other physico-chemical properties, which make aqueous electrolytes superior to other solvents, is the auto-proteolysis of the water molecules. The resulting hydronium or hydroxy-ions can exhibit extraordinary conductivities via the Grotthus mechanism [13]. These unique ionic transport properties are exceptional and lead to high conductivities, which result in low internal resistance of the cell. Moreover, aqueous electrolytes provide the battery with inherent safety as they are non-flammable, which is in contrast to most organic solvents. The high thermal capacity of water is also beneficial for simplified heat management (heat removal from the battery cell) when compared to Li-ion cells of hydrogen fuel cells.

### 2.2. Solubility and Strategies for Its Improvement

As mentioned above, the electrolyte consists of active material in the form of dissolved redox couples and a supporting electrolyte, which acts as an ionic charge carrier to maintain the charge balance during the conversion. The solubility of the active material determines the theoretical volumetric energy density in Wh/L. This is calculated with an adapted version of Faraday’s law of mass charge equivalent (1). This formula is only valid for symmetric systems of mixed electrolytes. Symmetric means that the amount of charges converted in the anodic and cathodic reaction is equal. As introduced earlier, the term mixed electrolytes means that there is only one electrolyte formulation in both tanks, containing the negative and positive electrode redox couple at the same time. To achieve this energy, two liters of electrolyte, one for the negative and one for the positive electrode reaction, are required. This is the reason why in the formula, the concentration is multiplied by a factor of ½:(1)$$\frac{E}{V}=Fz\frac{c}{2}U_{OCV}$$
where *E* = Energy in Wh; *V* = Volume; *c* = limiting concentration of active species in the electrolyte; *z* = number of transferred charge per unit (molecule/ion); *F* = Faraday’s constant; *U_OCV_* = Open Circuit Voltage;

For asymmetric systems, where the number of transferred charges per redox atom/molecule is not equal, a modified version of the formula is required (see Formula (2)), where the reduced volume due to the charges per ion/molecule are respected. Please note that in case of a symmetric flow battery, Formula (2) converts to (1):(2)$$\frac{E}{V}=Fz\frac{c}{\left(1+\frac{z_{1}}{z_{2}}\right)}U_{OCV}$$
where *E* = Energy in Wh; *V* = Volume; *c* = limiting concentration of active species in the electrolyte; *z* = number of transferred charges in cell/half-cell reaction; *z*_1_/*z*_2_ = ratio of the numbers of converted charges per unit (ion, atom, molecule) of active material (*z*_2_ > *z*_1_); *F* = Faraday’s constant; *U_OCV_* = Open Circuit Voltage

There are several ways to improve the solubility of active species. For inorganic as well as aqueous organic substances, usually the choice of counter ion has an influence on the solubility. The best known example is the increase in the solubility of [Fe(CN)_6_]^4–^ ions if sodium is exchanged partially with potassium, which results in an increase in the solubility of around 30% [14]. Even higher solubilities can be obtained for ammonium salts of ferrocyanide and ferricyanide (1.6 and 1.9 mol/L in water, respectively) [15]. Similar behavior has been observed for sulfonated anthraquinones, showing a significant effect of cation on solubility [16].

Another common approach used to increase the solubility in organic aqueous electrolytes is via functionalization of the redox moiety with solubilizing groups. A lot of functional groups, especially charged functional groups, have an impact on the solubility in aqueous media. The most prominent example can be found again in anthraquinones, as the pristine 9,10-anthraquinone molecule is almost insoluble. Functionalization, such as the di-sulfonated derivative (anthraquinone-2,7-disulfonic acid disodium salt), can significantly increase solubility to around 1 mol/L [17] at room temperature. Other substituents have even more striking effects on solubility. The groups affecting solubility can be subdivided into several categories. General trends in solubility for side-groups include:

**Negatively charged functional groups:** carboxyl-, phosphate-, phosphonic- < sulfate-, < sulfonic < sulfonimido-;

**Positively charged functional groups**: pyridinium- < Imidazolium- < tetraalkylammonium-;

**Non-charged functional groups:** cyano- < mercapto- < hydroxy- < morpholino- < polyether-groups. 

Especially for the charged groups, the solubilizing effect is highly dependent on the pH of the electrolyte and on the position in the molecule. Several principles can be taken into consideration, such as: (1) increasing the polarity of the molecule by introducing heteroatoms; (2) introducing charges, especially if they are sterically hindered; or (3) breaking the symmetry of the molecule. All these synthetic strategies can lead to increased solubility.

The third strategy is the application of a solubility promoter. These are organic molecules, which help polar molecules to stay in solutions. A typical example of a solubility promoter is the use of acetic acid to increase the solubility of usually poorly soluble phenothiazines up to molar concentrations [18].

In computational chemistry, new ways of predicting solubility with molecular dynamics exist [19,20]. These methods simulate the inner and outer coordination sphere of a molecule. The main problem today is still the difficulty of calculating the lattice energy of the solids. Nevertheless, these methods are becoming increasingly important for the layout of more soluble redox molecules or the discovery of a better solubility promoter.

### 2.3. Redox Potential

The value of the electrode potential is a quantitative measure of the redox activity of substances participating in the electrode reaction. The electrode potential is usually given by the Nernst Equation (3), which relates the half-cell potential to activities (concentrations):(3)$$E=E^{0}+\frac{RT}{ZF}ln\frac{\prod {\left(a_{ox,i}\right)}^{n_{i}}}{\prod {\left(a_{red,i}\right)}^{m_{i}}}$$
for an electrochemical reaction:(4)$$m_{1}a_{red,1}+m_{2}a_{red,2}+\cdots ↔ze^{-}+n_{1}a_{ox,1}+n_{2}a_{ox,2}+\cdots$$
where *E*^0^ = standard potential; *R* = gas constant, *T* = absolute temperature; *z* = number of transferred charges; *F* = Faraday’s constant; *a_ox,i_* = activity of oxidation reaction partners; *a_red,I_* = activity of the reduction reaction partners, *n*, *m* = stoichiometric factors.

The link between the cell potential and the concentrations is important for a flow battery, as the OCV (so-called open circuit volage, measured in zero current conditions) can be expressed by the difference of the electrode potentials of posilyte and negolyte. The OCV can provide useful information for the battery management system. The Nernst equation requires standard potentials. For aqueous organic molecules, it can be difficult to provide standard potentials. For the measurement of standard potentials next to the standard conditions (273.15 K and 1013.25 hPa) [21], the concentrations of the redox couple need to be set to unity. This is often not possible for organic molecules as very often, the molecules are present in an uncharged form. In principle, for these molecules, the concentration term in the Nernst equation has to be replaced by the fugacity of the molecule, which is in most cases a very inapproachable unit. The before mentioned reasons make the measurement of defined standard potentials very difficult. For this reason, mostly formal potentials (*E*°’) can be found in the literature. These are easily accessible from cyclic voltammetry but are usually more potential ranges than exact physico-chemical constants.

In technical flow battery electrolytes, the redox potential difference between the negolyte and the posilyte should ideally be higher than 1 V, as it is very difficult to achieve reasonable power densities with lower OCV. For illustration, the effect of the cell area-specific resistance (ASR) on the cell voltage and power density for discharge is shown in Figure 1, assuming a constant ohmic drop of 0.5 Ω cm^2^ up to 2 Ω cm^2^ and OCVs of 1.5, 1.0, and 0.5 V. From these curves, it can clearly be seen that the maximal power density drops with lower OCV. This is just a presentation of the best case, as additional non-linear voltage drops due to charge transfer and mass transport resistances are not included in these graphs. For example, typical ASR in cells for all-Vanadium cells at room temperature is between 1 and 2 Ω cm^2^ for thick felt electrode-based cells (3–5 mm thickness, approximately 10–30% compression rate) with flow through the electrolyte distribution down to 1 to 0.5 Ω cm^2^ for paper electrodes (usually 100–300 µm, usually 2–4 papers stacked) flow-field-type cells. Some high-power cells with super-thin membranes might have 0.3–0.4 Ω cm^2^, but with current materials and cell designs, the ASR will not get much lower in the near future due to the limits of the applicable membrane thickness and the resulting membrane conductivity. Thus, the chemistries with low OCV requires extra low ASR values of the cell, which is usually connected with higher costs for the stacks and/or the need for thinner membranes, which leads to enhanced cross-over of active species.

Fortunately, for aqueous organic redox couples, the redox potential of organic redox couples can be tuned by the substituents to the redox moiety. Especially, the redox potential of condensed aromatic heterocycles or quinone-type redox couples can be manipulated by different electron-withdrawing or electron-donating substituents in the condensed aromatic rings. The electron-withdrawing substituents can move the redox potential towards more positive values in the following trend: nitrato > cyanido > perfluormethyl > carboxy > chloride > fluoride (e.g., in phenazines) while electron-donating substituents, such as ethyl < methyl < thiol < hydroxy < ammino < triethylamino, may lower the redox potentials, e.g., in phenazines, if they are substituted at the outer positions of the adjacent aromatic rings [22]. These changes are also valid for quinones but not so pronounced as in the case of phenazines.

The second important influence on the redox potential of organic redox molecules is the pH value of the electrolyte during operation. The potentials can vary by multiples of 59 mV per pH unit, as long as protons or hydronium ions are active in the reaction. This means that during the reaction, either hydronium or hydroxy ions are consumed or produced. In a lot of cell reactions, the concentration of hydronium or hydroxy ions also changes in the molar range, due to electrode reactions and transport of the species through the membrane to equilibrate the charges transferred in the reaction. As a consequence, the half-cell voltage might shift significantly during the redox transition. Sometimes the requirements for chemical stability or reversibility for some organic redox couples over the whole course of charge-discharge operation with changing pH values are not easy to fulfil in an electrolyte formulation. An equilibrating concept, such as buffering, does not help significantly in highly concentrated technical electrolytes, as the hydronium/hydroxy ions can change in the molar range. Buffer capacities, on the other hand, can only equilibrate hydronium/hydroxy ion concentration changes in sub-molar dimensions. In some publications with low concentrated electrolyte, buffered solutions may be useful, but in a technical battery with concentrated electrolytes, the concept of pH buffering has only limited sense. This means that for most organic redox couples, the pH and redox potential changes during the redox transition have to be adjusted and the stability of the redox molecule over all pH transitions has to be ensured. It has to be noted that the organic molecules are slow-moving ions while hydronium or hydroxy ions move fast. This may also lead to charge accumulations and shifts in pH during battery operation, when a very selective ion-exchange membrane has been applied. Therefore, in the design of a technical electrolyte, the pH transition has to match the type of ion-exchange membrane to equilibrate pH changes during operation. Otherwise, gradients in pH might add up and may change the redox chemistry in an unfavored way.

### 2.4. Redox Kinetics

The cell electrode reactions in a flow battery should be fast with a high reaction rate using cheap electrode materials. Fast reactions lead to high electrode surface utilization, resulting in potentially high power densities, as long as fast transport of reaction media is ensured during the reaction. Thus, 3D electrodes are typically used for flow battery construction, based on fibrous materials (felts, cloths, or papers) with a significant electrode–electrolyte interface (which is the place of the conversion reaction), high electrical conductivity, sufficient chemical stability, and suitable texture (allowing flow of electrolyte through the electrode structure). In most of the RFB systems, carbonized polymeric materials are used, typically after their hydrophilization in an oven, although various chemical and physical methods have been reported for improvement of the electrode catalytic activity for a given reaction [23].

In most publications, the redox kinetics is measured with cyclic voltammetry on a glassy carbon planar disc electrode, due to the difficulty in correcting the effects of pore diffusion in voltammograms with porous electrodes [24]. However, kinetic data from organic molecules also have to be analyzed with some consideration. Organic redox couples are, in most cases, large and diffuse slowly. Often, these molecules tend to undergo different surface reactions depending on the potential, their local chemical environment, and local pH. In some cases, adsorption of the species on electrode surfaces makes the electrode kinetics look better than it actually is. As a result, this can lead to super short cycles in the subsequent flow-cell test, where only the charges adsorbed on the electrode are converted. A good example of adsorbed redox molecules can be seen in the voltammograms of ADQS in 0.5 M sulfuric acid in the article by Zhang et al. [25] Scientists should be very suspicious if the peak separation of the polarographic waves looks too narrow to be true (i.e., lower than the minimum for the Nernstian electrode process).

Additionally, the pH region in which the measurement is done can have a large influence on the reversibility of the reaction of some molecules. In the case of unsaturated heterocycles, e.g., pyrazines, quinoxalines, and phenazines, pH reactions, such as protonation, can have a severe influence on the stability of radicals due to the change of the aromaticity in these substances (see Figure 2) [26].

### 2.5. Chemical and Electrochemical Stability

Besides its low price, sufficient chemical and electrochemical stability is one of the key requirements for organic electroactive molecules. In principle, there are several mechanisms leading to the capacity fade of aqueous organic RFB, such as: electrolyte disbalance due to cross-over of active species through the membrane/separator and parasitic side reactions (water splitting or oxidation by the air), precipitation of active compound from the electrolyte, and (electro)chemical decomposition of active species. Some of these aspects are discussed in more detail in the following section.

#### 2.5.1. Stability of the Supporting Electrolyte–Electrolyte–Electrode Interphase

The electrochemical stability of the supporting electrolyte is always connected with the materials of the electrode and the material of the internal current collector (contact plate in a cell/bipolar plate in a stack). Aqueous electrolytes are limited by the narrow electrochemical window of water splitting. The corresponding thermodynamic potential difference has a value of 1.23 V, independent of pH, and can be extended up to 2.0 V considering the kinetic barriers of the hydrogen evolution and oxygen evolution reactions on electrodes, e.g., carbon electrodes. Within these borders, the cell can operate without side reactions of electrolyte decomposition (i.e., water splitting in case of aqueous electrolytes). The wide electrochemical operation window is the reason why carbon electrodes are mostly applied in flow battery applications. In some alkaline batteries, nickel or nickel alloys could also be applied. In some acidic systems, platinized titanium as well as cerment electrodes, such as dimensionally stable anodes (DSAs), are possible electrode materials, but these are known to catalyze oxygen evolution. The stability of these materials in the given aqueous electrolyte limits the electrochemical window in which the cell can operate.

#### 2.5.2. Electrochemical Stability of Active Species

In a recent review, Kwabi et al. compared the stability of five classes of aqueous redox-active organics and organometallics for which cycling lifetime results have been reported: quinones, viologens, aza-aromatics, iron coordination complexes, and nitroxide radicals [29]. The observed capacity fades, categorized from high” (>1%/day) to “extremely low” (≤0.02%/day), were found to be rather time-dominated than cycle-dominated and dependent on the electrolyte state of the charge and the concentration of redox species. Symmetric cell cycling with potential holds was suggested for the evaluation of capacity fades. This experimental approach, which was previously introduced by Goulet et al. [30], provides more accurate results when compared to standard galvanostatic charge-discharge cycling, without the effect of variation in the cell resistance and the permeation of the counter electrolyte.

Nucleophilic additions or substitution (Michael´s addition, gem-diol formation, substitution with water) are the most typical degradation mechanisms for posilyte active species, such as benzoquinone derivatives, with higher redox potentials due to the electron deficiency of their oxidized form, resulting in either loss of conjugation or a decrease in the reactant reduction potential and/or solubility [31]. The nucleophilic attack by hydroxyl ions has been identified as the main decomposition mechanism for nitroxyl radicals (TEMPO) and organometallic complexes, such as ferrocenes or ferrocyanide [32], which are typically used for posilyte in a neutral or alkaline environment. According to the presented studies, the highest stability of these types of active compounds should be achieved in neutral or only weakly alkaline conditions.

In case of negolyte redox active molecules, the decomposition mechanism typically involves disproportionation of the reduced form, leading to the formation of an electrochemically inactive compound, such as anthrone formation from reduced anthraquinones. This is particularly observed for acidic anthraquinone electrolytes. The mechanism has been well known for many years in aqueous organic batteries [33], but is also one of the main degradation routes in the oxidation of hydroquinone in the industrial anthraquinone process for the production of hydrogen peroxide [34]. Aziz’s group has lately developed a regeneration strategy for this degradation path by oxidizing anthrone with oxygen [35].

For bipyridine derivatives, such as methyl viologen (MV), disproportionation of the dimer of the partially reduced viologen cation radical (MV^+•^) leads to the formation of non-ionic and thus insoluble fully reduced (MV^0^), which deposits on the electrode surface, leading to the loss of active material and potential deactivation of the electrode. The addition of permanent solubilizing side chains has been reported to prevent the mutual interactions of the molecules (π-π stacking) via coulombic repulsions of the charged alkyl chains [36]. Moreover, the presence of even small oxygen traces can lead to a gradual increase of pH (via reaction of oxygen with partially reduced viologen), leading to dealkylation of the viologen molecule [37,38], which has been recently confirmed in [39]. Oxygen’s presence can also be detrimental for ferrocenes in the oxidized (i.e., charged) state, where bimolecular decomposition has also been reported [38]. The critical role of oxygen in decomposition mechanisms for neutral and near neutral organic RFB is a distinctive difference when compared to VRFB, where oxygen’s presence (due to insufficient tightness of the system) is only related to the losses of coulombic efficiency and SOC disbalancing.

### 2.6. Ionic Conductivity 

Ionic conductivity of the electrolyte is a parameter mostly influencing the ohmic resistance and therefore the discharge power density of an electrolyte system. The ionic conductivity is strongly connected with the ionic composition of the electrolyte. In Figure 3, the conductivities of the most used supporting salts are summarized. In most electrolytes, mid-concentrated strong acids or bases are used as electrolyte, as they offer the highest conductivity of all ionic substances. This is related to the extra conductivity via the Grotthus mechanism for the simple hydronium or hydroxy ion. If solubility is not limiting, usually the maximum conductivity of acids/alkalis can be achieved at around 4–5 mol/L hydronium/hydroxy ion concentration (see Figure 3), as at this concentration, the ratio between bulk water and water bound in the hydration sphere of ions is at its optimum. At higher concentrations, more water molecules are bound to the counter ion, restricting the mobility of the hydronium/hydroxy ions. Additionally, the viscosity of the solutions increases and has an effect on the ionic conductivities.

The ionic conductivity of all solvated ions in near neutral conditions, where no extra-conductivity mechanism is present, is directly related to the radius and charge of the hydrated ion and the viscosity of the medium. Usually, the conductivities are lower by a factor of 3 to 4 than in acids/alkali. Often, the conductivities are counter-intuitive, as, e.g., potassium ions have higher conductivities than, e.g., lithium or sodium ions, see Figure 3. This has to do with the size of the hydrated ion radius, which is higher for sodium than for potassium. For some molecular ions, such as ammonium ions, dissociation equilibria can also have an impact on the conductivity. The higher conductivity of ammonia salts (Figure 1) can be related to the extra conductivity of hydronium ions of the hydrolyzed ammonium ion.

The requirement for extremely high ionic conductivities in a technical flow battery can be highlighted in a comparison with a conventional lithium ion battery (LIB). In a flow battery, the electrolytes are flown through the porous electrodes with a thickness of some mm [48]. In some special flow field designs, the gap can be reduced to close to 1 mm or even below, but nevertheless, a certain gap is required to maintain the flow. In a LIB, on the other hand, the gap can be made as small as the separator itself. This means that LIB can achieve a higher geometrical surface area in a very small volume compared to RFB.

Although LIBs have relatively high volumetric power densities, the area-related power density is actually quite low. LIB cells with a high C-rate, e.g., 20C, hardly hit area-related current densities of 5 mA/cm^2^. In contrast, flow batteries provide 10–50 times higher values. Due to the necessity of the flow conditions, the electrode area cannot be packed as close as in conventional batteries, resulting in a lower active surface area, thus a lower power density per volume unit of an RFB stack. The implication of the necessary “gap” in flow battery cells means that flow battery chemistries require at least one-dimension higher area-related power densities [48] to achieve volumetric power densities in a similar range. This is the reason why the lowest possible ohmic cell resistances are a prerequisite for technical flow batteries. The increased power densities lead to smaller and thus cheaper stacks, which decreases the costs per kW. To achieve this, the ohmic drop on the electrolyte needs to be as low as possible and so far, only aqueous electrolytes can achieve the necessary ionic conductivities.

### 2.7. Dynamic Viscosity 

The viscosity of an electrolyte has a strong influence on the ionic conductivity but also on the pumping losses, i.e., energy losses due to electrolyte circulation in the battery. In Figure 4, the dynamic viscosities of the most common supporting electrolytes with. their concentrations are depicted.

The viscosities do not directly follow the trends of the conductivities. Usually, a lower viscosity would be expected to coincide with a higher conductivity, since the ion mobility is one important factor in conductivity. However, the results are in some cases counter-intuitive. Especially, pH-neutral sodium salts, for example, have a higher conductivity than the respective lithium salts, but the viscosities are often in the reverse order to the trend. This can be explained by the stronger binding of the second hydration shell resulting in a larger hydration radius of the lithium ion in the aqueous solution [53].

The addition of the redox couple to the supporting electrolyte changes the rheological properties of the electrolyte, especially for aqueous organic redox couples. Usually, all molecular electrolytes can be considered as Newtonian fluids; however, thixotropy has been observed for some quinone-based electrolytes at higher concentrations [17]. Additionally, some formulations propose polymeric electrolytes or slurries. These electrolyte systems have to be rheologically optimized for use in flow batteries. For these cases, the electrolytes should be at least shear-thinning to allow flow at low friction through the cell. In some cases, highly viscous fluids, such as slurries, might have an advantage, if they behave as non-Newtonian fluids. In this case, the slurries do not follow the slip condition and allow a better exchange of particles on the surface, which might allow higher power densities.

### 2.8. Permeation

Besides the degradation of active compounds, the capacity fade of the battery can also be caused by the permeation of active species through the membrane or separator, which is mutually separating the half-cells. In RFB, mostly homogeneous ion-exchange polymeric membranes (e.g., Nafion) are used, providing good selectivity, i.e., a compromise between high ionic conductivity and low permeability for electroactive species. The choice of a proper membrane strongly depends on the electrolyte properties, such as pH, oxidation strength, and charge of active species. For cationic active species, anion-exchange membranes with positively charged ion-exchange groups (quaternary ammonium or phosphonium groups) are typically used and vice versa, exploiting the coulombic repulsion between the membrane and active species. The selectivity of the homogeneous membrane depends on many parameters, such as the membrane thickness, ion-exchange capacity, and inner structure (width of hydrophilic transporting channels in hydrophobic matrix) [54], and it can be further enhanced by various methods, such as polymer blending [55,56] or inorganic-organic hybrid membranes [57]. Typically, a less corrosive environment of organic RFB electrolytes enables the use of cheaper non-fluorinated membranes [58]. In contrast to classical vanadium RFB, organic RFB typically uses different active compounds for negolyte and posilyte, thus the permeation not only decreases the coulombic efficiency but it may also cause irreversible capacity losses as the separation of permeated species from the electrolyte is often technically difficult and thus uneconomic. The cross-over of active compounds through the membrane can be significantly reduced by modification of the structure of active compounds, e.g., by addition of permanently charged side chains (also providing enhanced solubility) [36] or via their use in the form of water-soluble redox active polymers [59]. The significantly increased size of the electroactive polymers enables the use of cheap size-exclusion porous membranes, which can reduce the system investment costs. On the other hand, the remaining challenges originate from slower mass transport of redox polymers to the electrode surface and increased electrolyte viscosity [59,60,61,62].

One possible solution of the cross-over problem is the use of mixed electrolytes containing both active species mixed in both negative and positive electrolyte. Such an approach enables periodical re-mixing of the electrolytes, e.g., after certain capacity fade, which, in theory, should restore the initial composition of the electrolyte. On the other hand, mixed electrolytes typically provide lower solubilities and ionic conductivities and the utilization of the electrolyte is decreased to half when compared to the standard set-up [63].

Another way used to prevent the detrimental effect of active species permeation is the use of single bifunctional molecules providing two redox electrochemically reversible transitions of both negolyte and posilyte in the so-called symmetric cell concept [64,65]. However, only few industrially available chemistries can provide this challenging feature, such as indigo or some hydroxylated anthraquinone derivatives, which has been reported either in a flow [66] or non-flow cell configuration [67]. In past years, various bifunctional combi-molecules, where negolyte and posilyte moieties are covalently bonded, have been successfully designed, synthesized, and tested for RFB application using various combinations, such as bipyridinium-ferrocene [68], TEMPO-viologen [69], or TEMPO-phenazine [70]. Another synthetically less demanding way is based on ionic pairing of halide anion (active for posilyte) with organic, typically viologen-based, cation (active in negolyte) [71,72].

Due to the different coordination spheres of single- or double charged ions of the supporting electrolyte as well as the redox couple, the composition of free water in the electrolyte may change during the redox transition. The bound water molecules in relation to their charges have a strong influence on the osmotic balance of an electrolyte. The parameter used to describe these changes is the ionic strength. This parameter has an influence on the osmotic pressure of an electrolyte and as charges are produced and transferred via the separator/membrane to equilibrate the produced charges, the ionic strength of electrolytes changes during battery operation. The osmotic imbalance created by the changes in the redox transition leads to water transfer and changes in the electrolyte volume. This osmotic imbalance can have an influence on the overall composition of the electrolyte. To mitigate this, several measures are taken in a technical redox flow battery, such as mixing cycles (only possible with miscible chemistry), a hydraulic shunt [73], or some concepts used to osmotically equilibrate the electrolyte by the addition of salts after the first charge.

### 2.9. Effect of Temperature

The temperature of an electrolyte system not only has a strong influence on the viscosity and with this parameter on the ionic conductivity, but also on the redox reaction kinetics. For the charge passage at the electrode/electrolyte interphase, the temperature-dependent activation energy plays an important role on the velocity of this transition. Moreover, the diffusion coefficient decreases with decreasing temperatures, which also slows down the transport of active species towards the electrode surface. This is the reason why most flow batteries perform rather poorly under cold climate conditions. On the other hand, increased temperatures may facilitate unwanted phenomena, such as cross-over of active species through the ion exchange membrane, redox compound decomposition [29,74], and water splitting on the electrodes. Capacity fade mechanisms are explored in more detail in Section 2.5.

Temperature also has an important impact on the solubility of active species in electrolyte. For most substances, the solubility decreases with decreasing temperature, which may lead to precipitation of redox molecules and clogging of electrode felts and related damage in the cell. In some chemistries, especially with colloidal redox couples, higher temperatures can also have a severe influence on the solubility. In the case of colloidal chemistry, temperature-related precipitation is usually irreversible, as the colloidal nature cannot be recreated after precipitation.

### 2.10. Aquatic and Human Toxicity

Aqueous electrolytes with organic redox molecules are known to have additional advantages, such as non-flammability, non-volatility, and lower toxicity, compared to non-aqueous systems. Toxicity is an important consideration in chemical hazard and waste management. The concern about human health and eco-environment safety is a priority of systems deployment in residential communities. The organic molecules used as redox couples belong to a wide variety of functional chemical classes. From here, the chemical hazard of the discussed materials can vary a lot and it is often altered by the other components of the electrolyte (the pH value seems to be the most important here). The toxicity of chemicals is a very complex problem and can be analyzed via plenty of aspects [75]. The acute toxicity of studied organic chemical groups is sometimes mentioned by the authors of works dedicated to RFB. However, in our opinion, the toxicity of future commercial systems should be analyzed deeper, i.e., the migration speed in the environment, the lifetime, and the products of degradation are required to characterize the real harmfulness of a compound. The general effect of a group of compounds of the same type is directly determined by the chemical constitution. As in many other fields, the toxicity of the chemical compounds has been estimated using fast-developing in silico methods [76,77,78].

Quinones exist in nature, and they are present in several natural products, endogenous biochemicals, and drugs, and are generated through the metabolism of aromatic compounds. Quinones can also create a variety of hazardous effects including acute cytotoxicity, immunotoxicity, and carcinogenesis. In the context of toxicology, three chemical features of these chemicals are important, i.e., quinones are oxidants and electrophiles, and can in some cases become phototoxic by absorption of visible or ultraviolet light [79,80].

Biological and pharmacological activities as well as low toxicity towards mammals have been found for some pyridine and bipyridine derivatives [81]. They are also effective against pests and have been applied as commercial herbicide [82] for many years.

Pyrazines are currently authorized as flavors in food [83] and tetrazines have civil and military applications as explosive materials [84]. Tetrazines [85] and thiazines as well as many other heterocyclic compounds have received strong interest in the pharmaceutical research area because of their diverse pharmacological activities (such as anti-microbial, anti-mycobacterial, antifungal, antiviral, antitumor, antipsychotic, anti-inflammatory, etc.) [86]. Pyrazines, tetrazines, and thiazines do not belong to this group of toxic compounds.

Toxicity and eco-toxicity data are incomplete for many polar aromatic sulfonates. Concerning acute toxicity, it is observed that these derivatives are less toxic than the corresponding non-sulfonated compounds (sometimes strong detoxification is noticed in the case of highly toxic substances). However, prediction of the risk of the chronic toxicity for sulfonates is not possible [87].

Aromatic amines have been known to be carcinogenic to humans and animals and they can damage the nervous system or cause lung irritation [88]. However, a decrease in mutagenic activity can be achieved via sulfonation, carboxylation, deamination, or substitution of an ethyl alcohol for hydrogen in the amino groups.

Nitryl moiety has occasionally been introduced to redox-active organic compounds. Nitriles belong to highly toxic chemicals.

Abundant organic dyes exhibit redox activity in RFB. Some of them (such as indigo) are sourced from nature. Dyes are hazardous chemicals to human health in bigger amounts and their presence results in liver dysfunction, headache, and nausea [89]. The toxicity and carcinogenicity of azo dyes are related to the harmful products (aromatic amines) of the metabolic processes.

The physicochemical and electrochemical properties of organic electroactive compounds are easily and effectively modified by functional groups to improve cell performance. The highly tailorable properties of organic molecules are their big advantage, but such a process alters the toxicity, which is unknown and difficult to predict for the. obtained derivative. However, some similarities regarding the toxicological behavior have been observed related to the appearance of a given moiety in an organic molecule. The introduction of a halogen atom into organic compounds is generally associated with an increase in the toxicity of the obtained derivative [90,91]. Moreover, the bigger number of halogen substituents potentially increases the health hazards from the given analogue molecule. Additionally, the positions of the halogens relative to each other are also an important consideration. Very few halogenated compounds occur naturally. Most are produced synthetically in the laboratory or as a result of industrial processes. The toxicity of a derivative generally increases as the size of the halogen atom decreases. Moreover, halogen and hydroxy groups together give rise to highly poisonous substances considerably more effective than when present separately.

Organometallic compounds are present in nature, and they often have biological functions. In general, the toxicity of regular metal complexes corresponds to a low or moderate level and can be below the values typical for popular organic compounds. Organometallic compounds represent a special case for toxicity studies because the toxic effect may be caused not only by the organometallic compound itself but also by its components (metal ion or/and ligand molecules) or by the corresponding degradation products. In the case of metal, its toxicity cannot be regarded as a constant property, since it depends on the oxidation state, ligands, solubility, type of counterion, morphology of particles, and properties of the environment. Moreover, heavier metals are not necessarily more toxic than lighter metals, as it is commonly believed [75].

### 2.11. Abundance of Materials

The criticality of raw materials is indexed by the European Commission in two dimensions: (a) the criticality and (b) the supply risk due to limited or very local production capacities. In Table 3, the 30 critical raw materials from the EU 2020 list of critical materials are summarized and battery-related elements are highlighted. 

For VRFB, six materials are relevant, which include bauxit (aluminium end plate of stacks), flourspar (perfluorinated membranes), natural graphite (bipolar plates), phosphate rock or phosphorous (electrolyte additive), and vanadium (electrolyte). The last two raw materials are excluded if aqueous organic materials are used. The reduction of critical materials is one main advantage and incentive to utilize aqueous organic redox couples.

## 3. Family Tree 

In this review, the authors aim to provide an overview of the current applied redox motives, which are used to develop aqueous organic electrolytes. The other aim of the article is to provide some background for synthetic chemists in which direction they should focus their future work to design molecules, which could be applied in technical electrolytes for flow batteries. In Figure 5, the redox range of most of the reviewed chemistries are given and indicated by a color code, in which pH regime the certain chemistry operates. The first noticeable thing is that more redox couples are located in the negolyte space. This graph also clearly indicates that it is not easy to compile a fully organic redox couple with a voltage higher than 1 V, as you can only potentially combine redox couples from the same pH regime. These two observations illustrate the current situation well, in that in most technical electrolytes, the organic redox couples are usually paired with inorganic posilytes.

The potential of the energy density of the electrolytes can be derived from Figure 6. The volumetric capacities of half cells are usually in the range of around 80 Ah/L with a few exceptions. This relates to solubilities of 2 mol/L e^−^ equivalents. However, Figure 6 only indicates the potential of solubility and not the practically utilizable volumetric capacity. For technical electrolytes, the real energy density values in Wh/L are typically much lower; usually less than ¼ of the given value. This results in mean values below 20 Wh/L and even much lower.

The reasons are as follows:(1)Energy density can be limited by decreased solubility of the redox compound in the charged state (e.g., reduction of pyridiniums resulting in loss of its ionic nature).(2)No mixed electrolytes were considered. If this is considered, the energy density is divided by a factor of two.(3)In a mixed electrolyte, it is unlikely that solubility is not affected by the other redox species. This usually leads to lower solubilities of the active species and an increase in viscosity.(4)Solubility of equal amounts of conducting salt for each redox equivalent plus an excess to ensure base conductivity are required, further reducing the solubility of the active material.

Therefore, you seldom see technical energy densities of mixed electrolytes exceeding 20 Wh/L for aqueous organic electrolytes.

### 3.1. Negolytes

This article aims to provide a brief overview of potential aqueous organic redox couples and an indication of promising redox motives for further investigation. In Figure 7, the redox motives of negolytes are shown and discussed in the following sections.

#### 3.1.1. *p*-Quinones

From various chemistries that have been tested for RFB, the organic compounds from the quinone group were among the first candidates who brought the attention to the field of organic RFB.

##### Quinones (One Ring)

The redox potential of the 1,4-hydroquinone/quinone reaction can vary significantly due to the pH and substituents used and thus it has been reported for both negolytes and posilytes. In this section, we will discuss its suitability for negolytes. As mentioned in Section 2.5, quinone moieties are prone to several degradation mechanisms. The most well-known is π-π -stacking, which forms a solid addition complex called “quinhydrone”. In addition, other well-known degradation paths, such as Michael addition and loss of hydroxy functionality in acidic media (often referred to as “anthrone formation” in anthraquinones), create severe stability issues with quinones. In single ring quinones, these mechanisms are extremely likely if there are hydrogen atoms on one of 2,3 and/or 4,5 positions. Therefore, it was really surprising when the Technical University of Graz announced a stable battery operation with a methoxylated p-quinone modified from the artificial aroma vanillin, which can be extracted from wood waste [92]. The cycling of the yielded 2-methoxy-1,4-hydroquinone in a 0.5 mol/L phosphoric acid electrolyte in combination with o-quinone posilyte showed a capacity decay of up to a 10% of the theoretical capacity over 250 cycles. The authors claimed that the use of 0.5 mol/L phosphoric acid increased the chemical stability of the single-ring quinone against radical attack, while 1 mol/L phosphoric acid led to the known decay mechanism. It is highly unlikely that low concentrated phosphoric acid will be a good choice for a technical electrolyte, as the. proton capacity of 1 mol/L of hydronium ions of the first two proteolysis steps is far too low for usage in high concentrated solutions. It remains to be seen in future studies if this stabilization approach will be used in a technical battery, as phosphoric acid is not a cheap conducting salt.

Another publication used hydroxylated single ring quinones as a stabilized redox molecule. Hydroxylated derivatives of *p*-hydroquinone, namely 2,5-dihydroxy-1,4-benzoquinone (DHBQ), were also suggested for alkaline negolyte. When paired with ferrocyanide posilyte, the cell provided OCV of 1.2 V and relatively high capacity fade due to crossover of DHBQ to posilyte and its degradation via nucleophilic attack of the hydroxide ions on the unsubstituted carbon atoms of DHBQ. The improved stability was observed in less alkaline conditions (when pH was decreased from 14 to 12) and after structural modification (blocking the unsubstituted positions by phenyl groups or polymerization) [93].

##### Naphthoquinone (Two Rings)

Comparable attention has been paid to the naphthoquinone derivatives. In an alkaline environment, 2-hydroxy-1,4-naphthoquinone, the dye Lawsone derived from natural Henna (Lawsonia inermis), has been successfully coupled with 4-HO-TEMPO [94], providing relatively high cell voltage (1.3 V) and high capacity retention (99.992% per cycle). In another study, 2-hydroxy-3-carboxy-naphthaquinone was used in combination with potassium ferrocyanide, providing OCV of 1.02 V and stable operation with a capacity retention of 94.7% after 100 cycles [95]. The improved relevant properties of the original lawsone molecule can also be improved by its dimerization. RFB using alkaline bislawsone-based negolyte combined with a ferrocyanide posilyte provides comparable OCV (1.05 V) and can be operated at an increased current density of cycling (300 mA/cm^2^), see Figure 8 [96]. In a recent study, the selected hydroxylated naphthoquinones and anthraquinones were tested as negolytes in an alkaline environment, showing a better performance of the anthraquinone-based molecule, both in terms of the cell voltage and volumetric capacity [97].

##### Anthraquinones (Three Rings)

Anthraquinone has been frequently reported as a negolyte active species, at various electrolyte pH. In acid and neutral solutions, sulfonation is typically required [98,99] to increase the solubility of initially insoluble anthraquinone, while in an alkaline environment, hydroxylated [100] or carboxylated [101] substituents are preferred. The type and position of substituents can also be used to tailor the redox potential of the molecule. Er et al. [102] performed a computational screening study of quinone-based redox couples to study the effect of the molecular structure on the redox potential and solubility. In general, electron-donating substituents (such as –OH, –NH_2_, or –N(CH_3_)_2_) decrease the redox potential while electron-donating groups (such as –COOH, –CHO, –PO_3_H_2_, and –SO_3_H) increase the redox potential. The most significant shift in potential was observed when these groups were substituted for the quinone hydrogens adjacent to the ketone units. The most efficient solubilizing groups are –PO_3_H_2_, –SO_3_H, –NH_2_, and –N(CH_3_)_2_. Thorough experimental screening of various quinone derivatives was carried out by Wedege et al. including the study of electrochemical stability in RFB cells [17].

Anthraquinone-based negolytes can be combined with various posilytes. In acidic environments, Br_2_/HBr posilyte is most often used. A cell based on 2,7-anthraquinonedisulfonated acid (AQDS) in sulfuric acid and HBr posilyte provides high power densities due to fast electrode kinetics in both electrodes and low ohmic resistance [98,103]. However, the relatively low cell voltage (below 0.9 V), capacity losses due to bromine crossover to negolyte [104], and toxicological aspects of bromine represent the challenges to the technology. Additionally, insufficient stability of AQDS in its reduced form and at elevated temperatures has been reported, most probably due to anthrone formation [74]. AQDS-FeSO_4_ RFB has also been reported [105], providing low material costs but also very low cell OCV of 0.6 V. Recently, the hybridization of an AQDS-based negative half-cell with an oxygen electrode in the concept of a fuel cell and flow electrolyzer has been reported for stationary energy storage application, showing comparable performance to its vanadium analogue [106]. The OCV of AQDS-Br RFB can be increased to 1.3 V by a differential pH set-up where posilyte is operated at pH ~ 2, while AQDS negolyte is at pH ~ 8 [107].

In neutral environments, AQDS-based negolytes can be combined with ferrocyanide [108] or halogens (I_3_^−^/I^−^) [109]. The solubility of sulfonated quinones was found to be strongly affected by the type of counter ions, which is related to the ion solvation energy and lattice energy [16]. Ion exchange of AQDS from the sodium to ammonium form resulted in a three-fold increase of the solubility (reaching 1.9 mol/L in water) and corresponding volumetric capacity. The addition of short polyethylenglycol side chains on the anthraquinone molecule resulted in a dramatically increased solubility (up to 1.5 mol/L) and has been successfully paired with neutral ferrocyanide negolyte [110].

In alkaline environments, hydroxylated anthraquinone is combined with ferrocyanide; however, the stability of the ferrocyanide redox couple under basic conditions is questionable as (CN)^−^ ligand dissociation has been reported in strong basic conditions of pH 14 [32]. The stability of hydroxylated anthraquinone (2,6-DHAQ) can be further enhanced by its functionalization with highly alkali-soluble carboxylate terminal groups. The resulting 4,4′-((9,10-anthraquinone-2,6-diyl)dioxy)dibutyrate (2,6-DBEAQ) is six times more soluble than 2,6-DHAQ at pH 12 and provides a low capacity fade rate of <0.01%/day and <0.001%/cycle when tested in a symmetric cell [101].

#### 3.1.2. Pyridinium Compounds 

Nature makes vast use of the pyridinium redox couple, as a lot of redox reactions in cells rely on the NAD^+^/NADH/H^+^ redox couple. Like many biological reactions, it is quite hard to recreate these types of reactions with typical charge transfer on macroscopic electrodes. The redox reaction of a single pyridine moiety is usually irreversible and leads to adsorption or even electrochemical dimerization on the electrode [111,112] surface. Only one reported pyridinium-based redox couple has been published for use as negolyte in RFB [113].

Bipyridinium derivatives, on the other hand, are potential electroactive compounds mainly for near-neutral RFB negolytes due to their suitable redox potential, high solubility, and simple preparation. The two pyridinium molecules can be arranged in various mutual orientations: 2,2′, 3,3′, or 4,4′, which can significantly affect the relevant properties. The benchmark bipyridine for RFB application is methylviologen (so-called paraquat), which has been successfully used with various posilytes, such as TEMPO derivative [60,114,115] and $I_{3}^{-}/I^{-}$ [71]. Although bipyridines can undergo two consecutive one electron reductions of both quaternary nitrogen atoms, in case of methyl viologen, only the first redox step can be practically exploited for aqueous RFB due to the low solubility of fully reduced MV^0^ (non-ionic), which is deposited on the electrode surface [116]. Thus, modified viologens have been designed and tested containing permanent solubilizing side chains, such as 3-(trimethylammonio)propyl or 3-sulfonatopropyl [36,63,117], see Figure 9. Besides improved solubility of the reduced forms, the charged side-chains also sterically and electrostatically hinder the dimerization of reduced viologens. A similar strategy can also be used with hydroxy-alkyl [118] and phosphono-alkyl side chains [119]. Another interesting approach is based on assembling viologen molecules (from 3 to 15 units) into a regular structure for near neutral and acidic RFB negolyte, providing multi-electron transfer but also significant stability issues [120]. Recently, Burešová et al. synthesized and tested a broad series of bipyridine and bipyrimidine molecules to observe significant structure–property relationships in terms of electrochemical reversibility. In their subsequent study, this group investigated a series of over 20 original structurally varied azinium scaffolds, showing promising results particularly for 1,5-naphtyridine derivatives [26]. Bipyridine moiety can be covalently bonded with posilyte-active moieties, such as ferrocenes or nitroxyl radicals or ferrocenes, in so-called bifunctional molecules, as described in Section 2.5 dealing with crossover phenomena.

#### 3.1.3. Pyrazine Compounds

Pyrazine compounds are heterocyclic non-aromatic six-membered cyclic compounds with two nitrogen atoms in the ring. The redox transition proceeds in a two-electron transition. Like in quinones, pyrazines become aromatic when in a reduced form. This feature increases the reversibility of the redox transition, especially in condensed aromatic systems, such as quinoxalines or phenazines, where either the outer ring/rings or the inner ring will become aromatic in the redox transition, leading to a stabilizing effect. This feature can also stabilize aromatic intermediates, such as radical ions, which might appear in incomplete redox transitions. Like in quinones, π-π-stacking can lead to dimers (e.g., phenazinehydrine), which can build films on electrodes, which was observed by Inzel et al. with phenazines on gold electrodes in perchloric acid solution [121].

##### Pyrazines (One Ring)

The literature about pyrazine as a redox active molecule for flow batteries is scarce, although pyrazines offer a lot of advantages, such as a low molecular weight, high solubility (≈6.6 mol/L in H_2_O), and relatively negative redox potential. Voltametric studies in aqueous media were conducted by Klatt and Rouseff. In aqueous neutral to alkaline solution, pyrazines have three reduction peaks in the voltammogram. The last is irreversible, leading to degradation due to ring-opening. Below pH = 2, e.g., in 1 mol/L HClO_4_, pyrazine exhibits two reversible transitions [122] at ≈−0.232 and ≈−0.320 V vs. Ag/AgCl, while the kinetic of the second wave is probably accompanied by a follow-up-reaction or a slower electrode kinetic. It is highly likely that the observed chemical follow-up reaction may be attributed to π-π stacking, which was observed in phenazine redox compounds [121]. The reason for this change in reversibility between neutral and acidic conditions lies in the aromatic nature of the di-hydro-compound [123], which forms in highly acidified solution, while the single-protonated compound is aliphatic (see Figure 2).

A theoretical study on condensed pyrazines with other redox moieties has been published by Hjelms’ group at DTU [65] in an attempt to create a single molecule redox couple. In this paper, several two- and three-ring derivatives of condensed combinations of condensed pyrazine, pyridazine, para-, as well as ortho benzoquinone, 1,4-dithiine, and 1,4-dioxine units were calculated to achieve molecules with two redox transitions, high potential difference, and high solubility. After this theoretical study, the same group published an article in which they focused on single molecule pyrazines and quinoxalines in a voltametric study. Next to the different quinoxalines, they focused on six methylated as well as carboxylated pyrazine molecules [124] published in this issue. Unfortunately, they analyzed the first redox transition in slightly alkaline solutions (pH = 13) but not in the highly acidic region. They also found relatively sluggish kinetics of all the pyrazine moiety in alkaline conditions and preferred quinoxalines for further studies.

##### Quinoxalines (Two Rings)

Quinoxalines are quite soluble in aqueous media with up to 4.5 mol/L and offer relatively low redox potentials in the range of −0.02 V at pH = 7 and from −0.7 to −0.8 V vs. SHE in alkaline media. This makes these substances quite interesting for negolytes. The downside is the relatively low solubility in alkaline media (0.05 mol/L). Despite their potential, not many applications in flow batteries have been published so far. Quinoxalines as a negative electrode have been proposed by Milshtein et al. [125]. Thirty electrolyte compositions were screened by cyclic voltammetry, and the five promising compositions were identified as potential electrolytes. One of the first full cells was published by Leung et al. [5], who paired quinoxalines in 2 mol/L NaOH with MnO_2_ catalyst pasted on a nickel mesh as a reversible air electrode. The achieved current density was 7.5 mA/cm^2^. Increased overpotential on the negative side and gassing were observed during cycling of the cell. Another study was conducted by Hjelms’ group on substituted pyrazines and quinoxalines, already cited above [124]. In this study, they substituted quinolines in the sixth and seventh position with methyl or carboxylic groups and looked at the influences in the voltammogram in slightly alkaline conditions (pH = 13) with sodium chloride as conducting salt.

##### Phenazines (Three Rings)

Just like pyrazine and quinoxaline, the aromatic nature of the fully or completely non-protonated redox moiety stabilizes the redox transition either in acidic or highly alkaline electrolytes.

Phenazine is the redox motive of some natural redox mediators, such as vitamin B_2_—riboflavin, which is the redox-active moiety in the co-enzyme FAD in the respiratory chain. The basic moiety of the riboflavin is the pteridine ring, which is basically a phenazine, where one ring is exchanged with a pyrimidine ring. The first mentioning of phenazines as active components in an RFB system was reported by Schubert’s group in Jena by Winsberg et al. [70], which might be inspired by the article of Hong et al. from Seoul University about “biologically inspired pteridine redox centres” for organic batteries [126]. Winsberg combined the phenazine negolyte with a TEMPOL posilyte, but due to the reactivity of TEMPOL, molecule cross-over led to degradation. Therefore, they coupled the two molecules via a polyether bridge. This coupling has the advantage that the electrode is not deactivated by the reduced TEMPOL molecule anymore.

Another system was published by Hollas et al. [127] from PNNL in their article about bio-inspired negolytes for aqueous RFB in nature energy. In DFT analysis, derivates of the 7,8-dihydroxyphenazine were identified as the most promising substances. The 2-sulfate and carbonate compound were further investigated. They paired the sulfonated phenazine derivates with ferrocyanide in not too alkaline media (pH ≈ 14) and managed to dissolve 1.4 mol/L of the phenazine derivative. A capacity fade could be observed, which was related to the decay of ferrocyanide in the alkaline electrolyte. Their thesis was backed by NMR measurement to prove the chemical stability of the phenazine compound.

Wang et al. from Nanjing University used a similar system but changed the 7,8 hydroxy phenazine with a modified commercial dye Basic Red 5 [128], to build a cell with ferrocyanide in alkaline media. The same dye has also been used by Lai et al. [62], but in their approach, they paired it with cerium in methanesulfonic acid to achieve stable cell operation.

A more fundamental DFT investigation of phenazine looked at a complete range of substituents. De la Cruz et al. [22] from IMDEA looked first at the solvent effect on the redox potential of phenazine molecule vs. ferrocene because they explored membrane-less two-phase cells, which works in different solvents. This analysis was followed by an investigation of different substituents on position 1 and 2 of the phenazine ring.

In a relatively new study, Xu et al. combined a phenazine molecule with ferrocyanide in alkaline media [129]. They demonstrated that this chemistry could be operated stably at higher temperatures, such as 45 °C.

#### 3.1.4. Alloxan Compounds

Alloxans are cyclic condensates of urea with mesoxalic acid. The best-known relative of alloxan-type compounds is the barbituric acid, which is the mother compound of the earliest sleeping drugs, such as barbital. The compound alloxan itself is a mild oxidation agent with a formal potential of around −0.145 V (vs. SHE, at pH 7.4) [130], with the reduced form dialuric acid.

As the mesoxalic acid favors the acetal form, alloxans also have the tendency to form acetal bonds. It can also build half acetals with itself. The alkaline form of the half-acetal is well known under the name murexide or purple acid. Murexide itself has two reversible redox transitions [131] in aqueous media, but the solubility is far too low and the chemical stability in acidic media is also reduced. The alloxan moiety is part of the riboflavin molecule (see phenazines). The redox motive of the riboflavin can be regarded as a combination of an alloxan ring with a quinoline ring. The Aziz groups published an article about these riboflavin types of redox molecules. They paired the riboflavin-type molecule with ferrocyanide and cycled it for 400 cycles. Like for the phenazines, electron-donating groups at the 7 and 8 position of the phenazine ring helped to stabilize the molecule. The Aziz groups chose methoxy groups in this position and a better kinetic could be shown in a voltametric study.

#### 3.1.5. Thiazine Compounds 

Thiazine compounds have a very similar redox behavior to pyrazine compounds, and aromaticity can only be reached when the sulfur heteroatom is lacking electrons, that is to say, positively charged. This form needs some resonance stabilization by delocalization of a condensed ring system. This is the reason why thiazines are only used in the form of amine substituted condensed cycles, also known as thiazine dyes, such as methylene blue. Usually, the solubility of these dyes is quite limited, but with certain promoters, it can be boosted into the molar range, as it can be seen in the case of phenothiazines.

##### Thiazines (One Ring)

The authors could not find any relevant publication with mono-ring thiazine systems.

##### Naphthathiazines (Two Rings)

The authors could not find any relevant publication with dual-ring thiazine systems.

##### Phenothiazines (Three Rings)

The most prominent redox active thiazine compound is the phenothiazine derivative methylene blue. Methylene blue is a redox indicator with highly reversible kinetics at neutral pH. Unfortunately, the solubility of methylene blue is around 0.14 mol/L too low for an application as technical electrolyte but shows relatively high chemical stability [132,133]. More soluble methylene blue derivatives, such as azure A (1.5 mol/L in H_2_SO_4_), are also unable to achieve the solubility to meet the criteria for a technical electrolyte. One potential work around this problem is the application of solvation promoters. In a study of the University of Texas, acetic acid in combination with sulfuric acid was chosen to increase the solubility of methylene blue up to 1.5 mol/L, resulting in volumetric capacity of 71 Ah/L [18].

### 3.2. Posilytes

In Figure 10, the redox motives of posilytes are shown, which are discussed in the following sections.

#### 3.2.1. *o*-Quinones (One Ring)

The derivatives of benzoquinone provide sufficiently positive redox potentials to be used as posilyte active species [134]. Analogically to AQ, sulfonation of benzoquinone molecules can be used to enhance the solubility in aqueous media. Moreover, the electron-withdrawing effect of sulfonate groups moves the redox potential towards more positive values, which is beneficial for the cell OCV [102,135]. Among the first organic redox couples, 4,5-dihydroxybenzene-1,3-disulfonic acid (BQDS) was introduced as a promising posilyte material providing a relatively high redox potential (0.85 V vs. SHE) within the concept of all-quinone RFB together with AQ-based negolyte [136,137]. Low cell voltage and insufficient stability of BQDS due to Michael’s addition were the main problems of the technology. The addition of two methyl groups on benzoquinone resulted in improved stability of the resulting BQ derivative 3,6-dihydroxy-2,4-dimethylbenzenesulfonic acid (DHDMBS); however, the cell suffered from low OCV and relatively fast crossover of DHDMBS through the cation exchange membrane [138]. BQDS posilyte was also tested in the concept of a regenerative fuel cell using hydrogen oxidation on a gas diffusion negative electrode, providing a low cell OCV of 0.6 V and poor stability due to BQDS degradation [139].

#### 3.2.2. *p*-Quinone (One Ring)

The range of quinones is relatively wide. In certain combinations, p-quinones could also be posilytes. The substitution of the 2,3 and 5,6 position of a quinone with electron-withdrawing substituents will increase the redox potential. This can be demonstrated by, e.g., the redox potential of chloranil, a fully chlorinated quinone, which has a redox potential range of 0.0 V in alkaline up to 0.3 V vs. SHE (in acidic) [140]. Chloranil is not a good example of a quinone posilyte, as it has a quite low solubility and also exhibits a corrosive nature and high aquatic toxicity. However, the principle of implementing electron-drawn substituents can be demonstrated very well with this molecule. So far, sulfonated thioether-substituted p-quinone has been applied as posilytes by Gerken et al. [141]. The redox potential ranged between 0.6 and 0.9 V vs. SHE. As expected, fully substituted quinones showed remarkable chemical stability compared to the asymmetric substituted. Another strategy for improved stability is substituting p-quinone with protonated morpholino groups, which increases the solubility up to 2 mol/L (107 Ah/L) in water, and the formal redox potential to 0.89 V vs. SHE; however, the performance was found to be highly dependent on the pH and the electrolyte composition [142]. The further molecular tuning enabled the authors to enhance the electrochemical stability of the fully substituted derivates [143], see Figure 11.

#### 3.2.3. N-O∙ Radicals

The field of stable nitroxyl radicals and their applications has undergone evolution and expansion. The first known N-O∙ radical substance was Frémy’s salt. Frémy’s salt forms automatically by adding sodium sulfite to nitrites and oxidizes them either electrochemically or by oxidation agents, such as bromine. The colorful purple salt can be precipitated as potassium salt and can be regarded as one of the first known N-O∙ substances. The much more known TEMPO (e.g., 2,2,6,6-tetra-methylpiperidin-1-yl-oxyl) was a component of the first all-organic non-aqueous redox flow cell. This compound also demonstrated excellent redox activity in an aqueous system up to 10,000 cycles [59].

Alkyl nitroxides belong to the class of π radicals, but the distribution of the spine density depends strongly on the polarity of the solvent molecules. In water (and in all polar environments), the ionic resonance structure is more favorable from an energetic point of view.

Delocalization of the unpaired electron and steric restriction from the four methyl groups on adjacent carbons in TEMPO molecules provide high stability of the nitroxide radical originates.

The TEMPO free radical is oxidized to its corresponding oxoammonium salt TEMPO^+^ in a fast and reversible one-electron redox reaction. The redox potential for this system is quite positive, and this compound is used as the positive electrode active material.

Since TEMPO itself is insoluble in water, the hydrophilic group or ionic moiety must be introduced at the 4-position. Ionic substituent in redox active organic molecules leads to an increase in the electrical conductivity of solution and also allows a reduction of the amount of supporting electrolyte. TEMPO molecules with cationic and anionic moieties have been tested in RFB.

The electrochemical and physicochemical properties of TEMPO derivatives are highly dependent on their structure. The introduction of a hydroxyl group to the molecule (4-HOTEMPO) increased the solubility to 2.1 mol/L in water (0.5 mol/L in 1 mol/L NaCl_(aq)_) and delivered a stable capacity for 100 cycles with nearly 100% coulombic efficiency and working at high current densities [115]. A lower number of cycles for this derivative is related mainly to the fact that its molecules can cross over an anion exchange membrane and take part inside reactions with H^+^ or OH^−^ ions [144]. The other derivative, 4-trimethylammonium-TEMPO chloride (N^Me^-TEMPO), exhibits more stable and less cross-over features in the RFB system. Here, the additional hydrophilic and ionic trimethylammonium chloride group with electro-withdrawing features causes an increase in both solubility (3.2 mol/L in water and 2.3 mol/L in 1 mol/L NaCl_(aq)_) and an increase of the redox potential by 0.15 V [114]. The other cationic TEMPO derivative (glycidyl-trimethylammonium cation-grafted TEMPO; g+-TEMPO) was also tested in RFB [145]. The synthesis of this compound is less troublesome, but the solubility is a few times lower than the one of N^Me^-TEMPO. The next tested cationic derivative 4-[3-(trimethylammonio)propoxy]-2,2,6,6-tetramethylpiperidine-1-oxyl (TMAP-TEMPO) chloride exhibits visibly higher (4.62 mol/L) aqueous solubility and low cost of synthesis [146], contrary to other cationic derivatives, i.e., 1-methyl-imidazolium-functionalized TEMPO (im-TEMPO) [147].

Five other simple TEMPO derivatives have also been tested as redox-active compounds, and they have oxo- (O=), cyano- (CN-), amino- (-NH_2_), carboxy- (-COOH) [148], or sulfate (-SO_3_^−^) [149] functional groups at position 4, respectively. However, TEMPO molecules with cyano or carboxy moieties show poor solubility and stability as compared to the other compounds from this family group and they are not recommended as an effective redox material. In the case of TEMPO-OSO_3_, this compound may form zwitterion (TEMPO^+^-OSO_3_^−^), which is able to aggregate and precipitate [149]. Adequate supporting electrolyte must be added to avoid this side effect, which makes this system more cumbersome to use.

The TEMPO-family group of molecules is sensitive to pH due to reversible protonation of functional groups. Side reactions with OH^−^ or H^+^ ions in basic or acidic solution, respectively, are observed [144]. In redox flow batteries, these compounds are utilized at neutral pH in the presence of a simple salt, such as NaCl [115].

It is worth emphasizing that this group of chemicals is distinguished by a small influence of supporting electrolytes or pH of the solution (in the rage of 1–10) on its electrochemical parameters [148]. Moreover, the difference in the salting out effect caused by supporting electrolytes is weak and does not differ visibly for the most popular salts.

A water-soluble bipolar organic compound has been presented recently as a new type of redox-active organic material [70]. This innovative approach is about the connection of a positive electrode active material (able to be oxidized) with a negative electrode active material (able to be reduced). The so-called combi-molecule utilized by Winsberg in RFB consists of two TEMPO subunits and one phenazine subunit covalently bound via TEG linkers. The capacity of the cell is not irreversibly affected by the cross-over of the active material into the opposite half-cell, which is a big advantage of such a system. The TEMPO unit takes part in a reversible redox reaction whereas the phenazine unit presents a quasi-reversible redox reaction. The combination of a viologen unit and TEMPO molecule also gives the redox-active material (VIOTEMP) [69]. However, this compound shows some disadvantages, i.e., the solubility differs a lot for different oxidation states and the electrolyte must be supported by a buffer to avoid huge change of the pH value during operation of the cell and to prevent side reactions.

As it is known, the bigger the size of the molecule, the smaller the chance for cross-over through a membrane. Following this idea, polymer (TEMPO-based; 22,200–33,700 g/mol) was synthesized and successfully utilized as a redox active compound in aqueous electrolyte [59,114], self-assembling into micellar structures in organic carbonate-based electrolytes [149]. The redox reactions during the work of the cell led, however, to some changes in the micellar structure. The viscosity, rheological, and transport properties of such electrolytes can take different values (not always convenient for RFB) and must be taken into consideration during implementation.

The reduction potential of TEMPO including redox-active materials ranges from 0.82 to 1.1 V (vs. SHE) and increases (for moieties introduced at the 4-position in TEMPO molecule and polymolecule “composites”) in the following order: -COOH < -O(CH_2_ )_3_N^+^(CH_3_)_3_ ≅ -OH < -OCH_2_CH(OH)CH_2_N^+^(CH_3_)_3_ < combi-molecule < -OSO_3_^-^ < -CN < -NH_2_ < polymer < =O < -N^+^(CH_3_)_3_.

*N*-oxyl compounds represent a versatile class of organic radical reagents with unique reactivity. The redox potential of *N*-oxyl radicals is associated with the O–H bond dissociation energy. Phthalimide *N*-oxyl (PINO) is also a widely used member of this group and is obtained via oxidation of stable N- hydroxyphthalimide (NHPI). PINO tends to self-decompose rapidly in pure aqueous acid solution [150], so the NHPI redox couple is not reversible in this condition. A reversible NHPI positive electrode compound with the potential of +1.30 V (vs. SHE) was achieved in a semi-aqueous redox cell by Tian and coworkers [151]. A polymeric additive is required in such a system for stabilizing the formed micro-heterogenous electrolyte. The optimal volumetric ratio of used solvents was found to be 50/50 (water/acetonitrile). It is worth noting that N-Oxyl compounds are also applied as a mediator in electrochemical oxidation reactions.

#### 3.2.4. Metal Complex–Low Spin

The ligation of chelate metal complexes may have a strong influence on the redox potential. Especially, N-functional ligands can form low-spin complexes of transition metals. These complexes can shift the redox potential of metal centers by several units. One of the most prominent examples is the cobalt-hexamine complex, which can stabilize the Co(II)/Co(III) transition in aqueous media, which would otherwise be out of the range of the stability region of aqueous electrolytes. The complexation of transition metals, such as iron, cobalt, and nickel, with N-functional chelate ligands is not widely used in flow batteries. The one prominent exception is ferricyanide or other cyanide compounds, which is also a low-spin complex of iron. In fact, ferricyanide is one of the most abundant redox active materials, especially when used in alkaline RFBs. Like the heavier brother element ruthenium, e.g., the ruthenium-phenanthroline complexes, all N-ligated iron complexes possess a pronounced photochemistry as well. This is why ferricyanide should be kept away from light. Usually, photochemistry is accompanied by photoreduction and ligand oxidation. In the case of ferricyanide, this leads to dicyan formation [152] and hydrolysis of the complex to either Prussian blue or in most cases to insoluble iron(II)/iron(III) hydroxides. The best way to stabilize these electrolytes would be the addition of cyanide to maintain complex equilibrium always on the complexed side. However, since cyanides are highly toxic and during cell operation alkaline conditions have to be maintained all the time to avoid the formation of toxic prussic acid, this stabilization method is never used in practice.

There are other potential low-spin complexes, which could be interesting. For example, porphyrin analogs could be an interesting substance class. Porphyrins are one of the most abundant redox couples in nature and potentials can vary from oxidizing to reducing conditions, depending on the central element and the nature of the surrounding of the redox center. Unfortunately, porphyrins exhibit a high molar mass and a rather low aquatic solubility. Some more simple planar low-spin complexes, such as nickel-cyclam, could be interesting positive electrode materials, as they can have high redox potentials (≈0.8 V vs. SHE, pH = 7) and the unfunctionalized material already has a solubility of 0.25 mol/L. There are not many studies on these complexes [153,154], and none for aqueous electrolytes. In non-aqueous media, the nickel cyclam complex possesses two redox transitions. Unfortunately, in aqueous media (pH = 7), the second transition at lower potentials seems to be irreversible [155]. However, there is quite a potential in N-ligated transition metal complexes for new posilytes, which has not been fully explored.

#### 3.2.5. Metal Complex–High-Spin (Ferrocenes)

The first scientific interest in this chemistry came with a press release of Lockhead Martins about upscaling a flow battery chemistry based on metal ligands. Since Lockhead Martin has patented electrolytes based on ferricyanide- (low-spin complex) and titanium-based high-spin complexes, such as Ti-catechol complexes [156], it is believed that their battery chemistry is based on these patents. However, the exact chemistry is a company secret and remains unknown, as no peer-reviewed paper has been published on the chemistry so far.

High-spin metal-organic compounds are seldom mentioned as redox active compounds in flow batteries in peer-reviewed literature. The one exception is the widely used cyclopentadienyl complexes, such as ferrocene derivates. Apart from this group, chelate metal ion complexes are seldom applied in aqueous chemistry. The second most cited literature is on the chelate complexes of EDTA derivatives. Examples include the all-chromium EDTA [157,158] complexes, the EPTA complexes also from chromium (EPTA = 1,3-propanediamine tetraacetate) for ferricyanide chromium RFB, or the diethylenetriaminepentaacetic acid (DTPA) complexes of vanadium, chromium, or cerium. Usually, the chemical stability of these complexes is not sufficient for regular flow cell operation. Apart from this, some unconventional complexes have been applied as flow cell electrolyte, e.g., biomimetic cobalt-sulfur complexes [159] or iron-triethanolamine complexes [160].

The most common group of high-spin metal organic redox couples is the ferrocene complexes. The use of ferrocenes for aqueous RFB posilyte was introduced in 2016 by the group of Leo Liu, which is very active in the development of new chemistries for neutral aqueous RFB [161]. Their ferrocene derivative ferrocenylmethyl) trimethylammonium chloride provides a relatively high redox potential (0.6 V vs. SHE) and can be solubilized by various substituents, resulting in high solubilities (4.0 mol/L in water). The neutral cell chemistry combining this ferrocene with methylviologen provided a high theoretical energy density (45.5 Wh/L) and excellent cycling performance from 40 to 100 mA/cm^2^. Unprecedented cycling stability was reported for bis((3-trimethylammonio)propyl)ferrocene dichloride posilyte with the bis(3-trimethylammonio)propyl viologen tetrachloride negolyte, showing 99.9943%/cycle and 99.90%/day at 1.3 mol/L reactant concentration [38].

The effect of the substituent chain lengths on cationic ferrocene molecules was thoroughly studied in [162], showing that both the solubility and stability of the ferrocene derivative is related to the localized LUMO density on ferrocene. The most stable BQH−Fc, which has the lowest LUMO density on Fe, provided a high capacity retention rate of 99.993% h^−1^ at 1.5 mol/L concentration.

Anionic sulfonated ferrocene, ferrocene bis(propyl sodiumsulfite), was recently reported [163], showing high solubility (2.5 mol/L in water) and redox potential of 0.23 V vs. SHE. The posilyte was tested with Zn-based negolyte, providing volumetric capacity of 40.2 Ah/L, OCV around 1.2 V, and 97.5% capacity retention after 1000 cycles. The stability of another anionic ferrocene 1,1′-bis(sulfonate)ferrocene dianion disodium salt (redox potential of 0.85 V vs. SHE) was tested in various supporting electrolytes and ferrocene decomposition due to nucleophilic attack by sulfate and acetate was observed, while good stability was observed in the presence of nitrate ions [164]. The electrochemical properties of all-ferrocene salts consisting of ferrocene cation and ferrocene anion were tested by Schrage et al. [164] as a bifunctional ionic electrolyte for the symmetric RFB set-up.

Ferrocyanide is probably the most frequently used posilyte active compound in the organic aqueous redox flow battery electrolyte, due to its low price and stability in neutral and alkaline conditions. The solubility of the posilyte is limited by the reduced state [Fe(CN)_6_]^4−^ and it can be significantly increased by ion exchange to the ammonium form, providing solubilities above 1.6 mol/L (43 Ah/L) while only 0.56 =ad 0.76 mol/L for sodium and potassium forms, respectively [15]. Previously, the sufficient stability of ferrocyanide-based electrolytes upon cycling in a symmetric cell has been reported for neutral and near neutral environments, while fast capacity fade was observed in alkaline conditions (1 mol/L KOH) due to the nucleophilic attack by OH^−^ ions [32]. Recently, the stability of ferrocyanide in alkaline electrolyte was advocated by Páez et al. [165], arguing that the observed capacity fade is related to electrolyte SOC disbalance due to an increased rate of parasitic oxygen evolution under the tested conditions. The energy density of ferrocyanide posilyte can be increased via use of a so-called capacity booster, i.e., high energy density solid stored in the electrolyte tank where chemical redox reactions between the solid and electrolyte active species (redox mediator) proceed. For this purpose, commercially available materials based on Ni(OH)_2_ have been successfully tested [166].

Besides well-known cyanide-based complexes, various other ligands have been tested for application. Howthorne et al. prepared and tested 7 various ligands for posilyte iron complexes showing promising properties for glycine iron, providing 0.5 mol/L solution at pH of 2 and redox potential of 0.690 V vs. SHE [167]. Iron azamacrocyclic complexes containing imidazole moieties were reported as a pH-tunable posilyte and negolyte for RFB application [168].

Iron complexes have also been tested for aqueous RFB negolytes. Triethanolamine iron complex (Fe-TEA) low redox potential (−0.86 V vs. SHE in alkaline environment) was used together with Br-based posilyte, showing impressively high cell OCV above 1.8 V; however, the stability was found to be strongly related to the concentration of Fe-TEOA in negolyte. In another study, Fe-TEA was coupled with ferrocyanide posilyte in so-called all-soluble all-iron RFB, providing relatively high OCV (1.34 V) and discharge power densities (160 mW/cm^2^); however, the cross-over of the free TEA ligand and poisoning of the cation-exchange membrane were identified as major challenges for further development [160,169]. According to Shin et al. [170], the poor stability of Fe-TEA results in deposition of metallic iron on the negative felt electrode. They addressed this issue by using more stable ligand 3-[bis (2-hydroxyethyl) amino]-2-hydroxypropanesulfonic acid (DIPSO), improving the cycling stability of RFB using ferrocyanide posilyte.

## 4. Customization of Components Used in Non-Aqueous Systems

Aqueous and non-aqueous electrolytes are two competitional systems with potential application in RFB. A big disadvantage of aqueous electrolytes is that many redox active organic components are insoluble in water, or the solubility is much lower than in other solvents. Additionally, the stabilization of radicals is often better in non-aqueous media. However, the synthesis of hydrophilic derivatives from hydrophobic compounds is one of the basic fields of organic chemistry interest. We would like to present some different redox motives, which in our opinion could be potentially applied in aqueous RFB after chemical modification.

### 4.1. Sulfur Compounds 

Sulfur compounds have an interesting electrochemistry, which can work in aqueous and non-aqueous electrolytes. The best examples for a low potential redox couple based on sulfur chemistry may be the iron-sulfur complexes in the respiratory chain in living cells. The most well-known human adaption to this redox couples is the two Roussin salts, which were described for the first time by François-Zacharie Roussin in the 19th century. Unfortunately, for a flow battery application, these salts are highly insoluble. Additionally, other attempts to apply this redox motive in a flow battery only could be achieved in a non-aqueous battery [171]. Sulfur-organic or sulfur complex chemistry has great potential for reversible redox chemistry.

Tetrathiafulvalene (TTF) exists in three different stable oxidation states and belongs to a small group of organic molecules, which can be reversibly oxidized or reduced without chemical side reactions [172,173]. Such unusual high stability of this compound should be used effectively in aqueous systems by introducing the hydrophilic or ionic moiety. In the neutral state, TTF has strong π-donor features, but an oxidation process decreases the π-donating effect. Both redox-transitions (to a radical cation or dication species) are fully reversible and tetrathiafulvalene and all oxidation states are not sensitive to the presence of air or moisture [172].

### 4.2. N-heterocycles

1,2,4,5,-Tetrazine (known as s-tetrazine) is a small, high nitrogen content heterocycle with very specific physico-chemical properties usable for several types of applications. Our interest in s-tetrazine derivatives focuses on their redox properties and reactivity. Among the C-N heterocycles, s-tetrazine constitutes the most electron-deficient six-membered aromatic ring and it can display reversible reduction by accepting one electron to form stable anion radicals but only in aprotic systems. From here, the compound modification should give high stability for both oxidation states in aqueous solution. Many tetrazine derivatives can also accept the second electron, but this process is in most cases not electrochemically reversible in standard conditions [174,175], while other studies have shown in very few cases a reversible electrochemistry in phosphate buffer [176]. However, more extended studies are required. A first step in this direction is the use of dihydrophtalazines, which very much resembles fusion of half of a pyridazine ring with a quinone moiety as a potential posilyte by the Schröder group in Gießen (now Braunschweig) University [177].

### 4.3. Stable Organic Radicals

Another unexploited field is N-radicals in the bridgehead position, such as the N in 1,4-diazabicyclo[2.2.2]octane, often called DABCO or TEDA (trimethylenediamine). Electrochemical oxidation in aqueous media is possible and since the rule of Bredt, these radicals are extraordinary stable [163]. DABCO is also a very cheap and soluble precursor. The problem with DABCO is the change in the acidity of the α-hydrogen next to the radical. Honarmand et al. found that electrochemical oxidation of DABCO led to abstraction of the α-proton and degradation in aqueous media [178]. Synthetic chemists may exploit this stabilization mechanism in other molecules in the future.

## 5. Outlook

In this article, a summary of different organic redox motives is provided and the authors attempted to structure the different motives to provide an overview about the possibilities for their use in technical electrolytes. Especially for synthetic chemists, this overview paired with additional information might be helpful to identify new synthesis strategies for useful redox active molecules, which are suitable in commercial batteries. Changes of the side groups on the main redox motive can influence physicochemical parameters, such as solubility, redox potential, and chemical stability, which can be exploited to design next-generation organic electrolytes with higher chemical stability, better reversibility, and higher energy as well as power density. Nevertheless, the most important step in the development of technical electrolytes for RFB is the design of the cheapest electrolyte. The important side note here is that this does not necessarily mean synthesis of the molecule with the lowest CAPEX. In energy storage applications, the cheapest electrolyte in particular needs to have the longest lifetime. Even the cheapest electrolyte can become expensive if it has to be changed every week or if it damages the electrode or membrane irreversibly. Therefore, technical electrolytes have to be designed with great care to achieve longevity of a battery.

The main advantage of organic electrolyte is definitely the availability of the components and the possibility to tailor the solubility and electrochemistry. The biggest drawback of organic electrolytes is the lack of recovery, once the redox moiety has been lost due to side reactions. Additionally, film formation due to adsorption and polymerization is one degradation phenomena that can be observed quite often. This makes the maintenance of electrolyte more difficult in such systems.

All future RFB chemistries must be benchmarked against the standard of the all-vanadium battery (VRFB). At the moment, only very few organic systems can compete with VRFB. Most systems have a lower cell voltage and lower solubility of the active species. This leads to higher system costs due to a higher number of cells in a stack, bigger tanks, and less efficiency due to higher pumping energy losses due to pumping of lower concentrated electrolytes to achieve similar reaction rates.

Some organic chemistries also have a lack of rebalancing capabilities for redox cross-over and osmotic imbalance. Another disadvantage is the lack of robustness of some systems. For example, a lot of technical flow battery developers regard miscibility of the chemistries as a prerequisite for a long-lasting battery chemistry. Unfortunately, a lot of chemistries do not allow mixing, as the different potentials of the other half-cell may lead to adsorption, polymerization, or even decomposition of active species.

A lack of recyclability might also have an impact in the future, as increasingly more sustainable solutions for batteries are required. Often, the favorable environmental aspects of organic electrolytes in contrast to metal-based ones are highlighted, but this argument is, at least for most current organic electrolytes, very questionable. Even the most toxic oxidation state of vanadium, which is the oxidation state V(V), does not have a severe environmental influence, as it becomes instantly insoluble, once it contacts ground water. This means that the aquatic toxicity of vanadium is in fact fairly moderate. On the other hand, a lot of the organic redox couples have higher toxicologic impacts on aquatic bodies. Therefore, the authors advise checking the aquatic toxicity before making such claims. Unfortunately, for a lot of substance classes, this information is rather incomplete or not available at all due to their minor application as industrial chemistries.

The targets for the battery chemistry are high. The levelized costs of storage were defined by the SET-Plan with 0.05 €*cycle/kWh by 2030. This means that for CAPEX costs of a 500€/kWh storage system (complete battery system!), the storage system has to last 10,000 cycles. The problem for long-duration storage is that these systems have a systematic disadvantage against short-duration storages, as 10,000 cycles only takes 105 days with a 15-min cycle operation but exceeds the duration of 27 years, if a cycles takes one day. Such a long duration is difficult to achieve in every product but also has implications on the financial side of such storage devices, as writing-off times in companies is 4–20 years for installations, depending on the investment. Therefore, these targets are not always helpful for the developer, even though these numbers remain in the flow battery community as the holy grail of battery storage.

Before a potential scale-up, the battery chemistry requires thorough investigations. The criteria for the users should be: (1) energy density, (2) power density, (3) lifetime (cycle life with rated power + additional exhaustive cycling test with I(U) charging to achieve highest possible SOC), (4) capacity recovery mechanism (reversing osmotic imbalance as well as irreversible redox shifts, reversing degradation of redox active species), and (5) costs of base materials.

If we look at probably, currently, the most low-cost organic-organometallic RFB system using anthraquinone-based negolyte in combination with ferrocyanide posilyte, with the costs of electrolyte active species produced on an industrial scale (10–50 EUR/kg for AQDS), we can hardly get below 60–80 EUR/kWh, which are the current limiting costs of vanadium in VRFB electrolytes. In fact, with AQDS at costs of 10 EUR/kg and cell voltage below 1.0 V, 100 EUR/kWh is exceeded only when considering a negolyte active compound. These systems may not offer comparable energy densities to VRFB but at least provide a better environmental impact than many other chemistries. Moreover, the reduced energy density originating from the lower cell voltage and solubility results in a further increase of the capital costs (bigger tanks and stacks) as well as operating costs (higher pumping losses). Thus, the overall economy of the storage needs to be considered within RFB development and should also be reflected in the cost targets for electrolyte active materials.

## Figures and Tables

**Figure 1 molecules-27-00560-f001:**
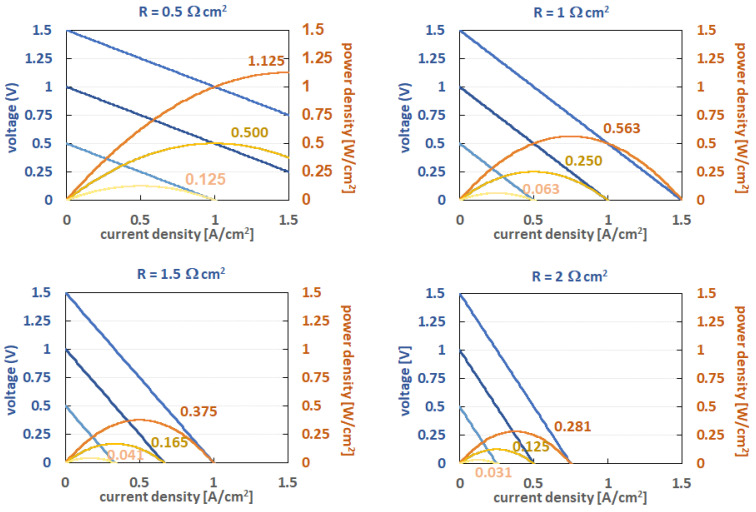
Estimation of the upper power density limits by comparing the ohmic power drop of different OCV (1.5, 1.0, and 0.5 V) with different ASR (0.5, 1.0, 1.5, 2.0 Ω cm^2^) assuming a constant voltage drop by applying ohmic law. The maximum power densities are highlighted.

**Figure 2 molecules-27-00560-f002:**
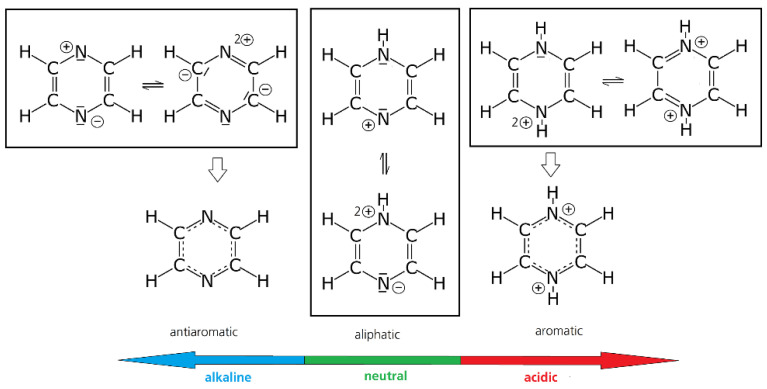
Example of the influence of pH on chemical stability. Tautomeric structures of a non-protonated (alkaline environment), single protonated (neutral environment), and double protonated (acidic environment) pyrazine ring and the related changes of the anti-aromatic, aliphatic, and aromatic state [27] (it should be mentioned that the anti-aromatic nature of unprotonated pyrazines is still under discussion [28]).

**Figure 3 molecules-27-00560-f003:**
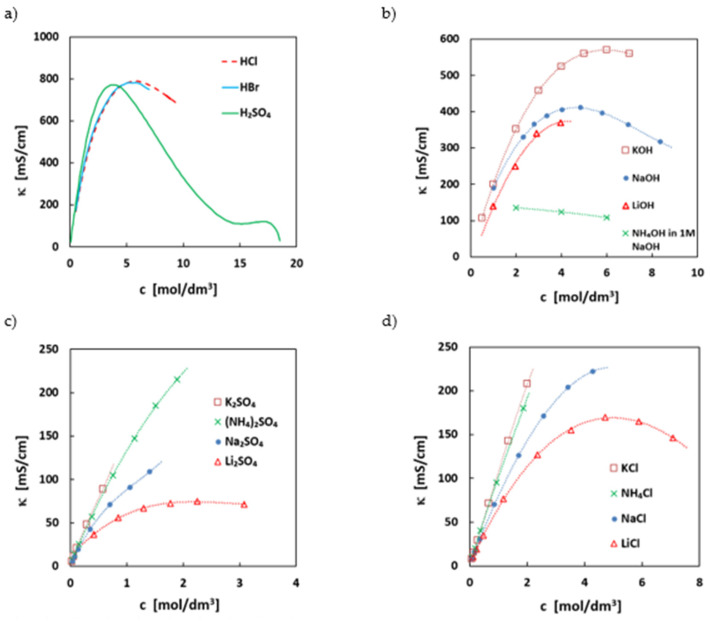
Variation of the specific conductance (κ) with the concentration (c) for some electrolyte solutions used in RFB at 293.15 or 298.15 K. (**a**) sulfuric acid [40], hydrochloric acid [41], and hydrobromic acid [41]; (**b**) sodium hydroxide (●) [42], potassium hydroxide (**□**) [43]; lithium hydroxide (∆) [44], and ammonium hydroxide in 1 mol/L aqueous NaOH (x) [45]; (**c**) sodium sulfate (●) [46], potassium sulfate (**□**) [46], lithium sulfate (∆) [47], and ammonium sulfate (x) [46]; (**d**) sodium chloride (●) [46], potassium chloride (**□**) [46], lithium chloride (∆) [46], and ammonium chloride (x) [46].

**Figure 4 molecules-27-00560-f004:**
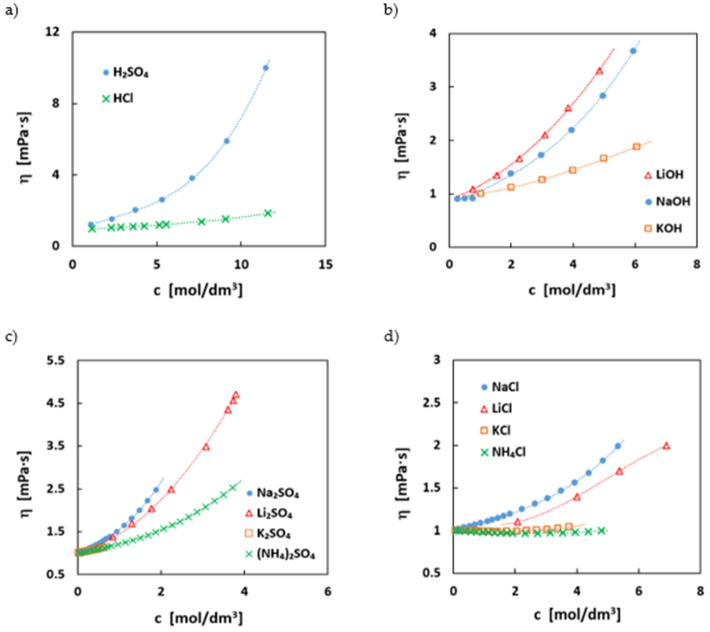
Variation of the viscosity (η) with the concentration (c) for some electrolyte solutions used in RFB at 298.15 K: (**a**) sulfuric acid (●) [49] and hydrochloride (x) [50]; (**b**) sodium hydroxide (●) [51]: potassium hydroxide (**□**) [51] and lithium hydroxide (∆) [51]; (**c**) sodium sulfate (●) [46], potassium sulfate (**□**) [46], lithium sulfate (∆) [47], and ammonium sulfate (x) [46]; (**d**) sodium chloride (●) [46], potassium chloride (**□**) [46]; lithium chloride (∆) [52], and ammonium chloride (x) [46].

**Figure 5 molecules-27-00560-f005:**
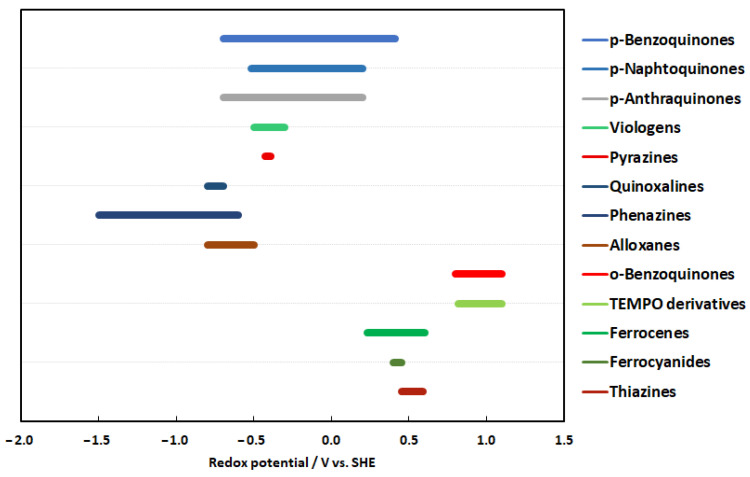
Redox potential ranges vs. SHE of redox couples discussed in this article. The color code of the bars indicates the pH regimes the redox potentials were measured in, red for acidic pH range, green for neutral, and blue for alkaline pH range, grey means that alkaline as well as acidic electrolytes were considered.

**Figure 6 molecules-27-00560-f006:**
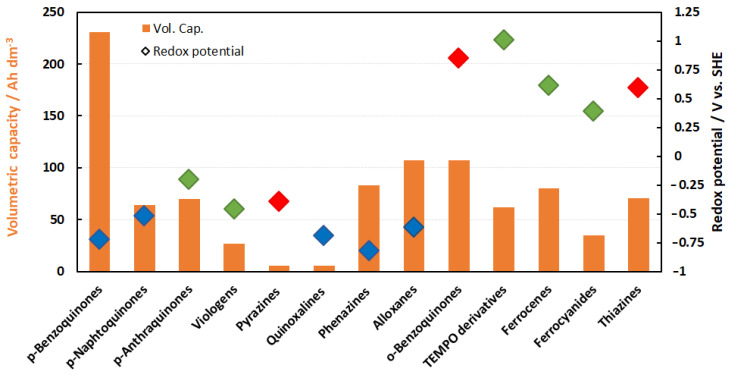
Redox potential ranges of redox couples discussed in this article. The color code of the bars indicates the pH regimes the redox potentials were measured, red for acidic pH range, green for neutral, and blue for alkaline pH range, grey means that alkaline as well as acidic electrolytes were considered.

**Figure 7 molecules-27-00560-f007:**
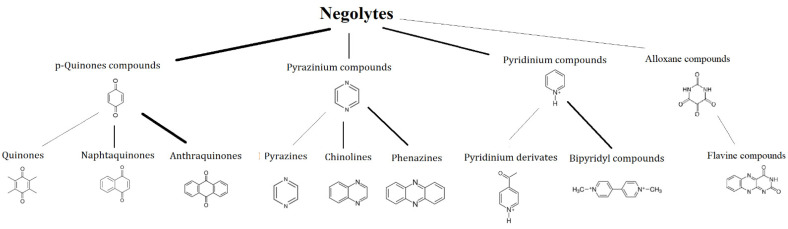
Family tree of the chemical motives for organic redox-active molecules that could act as negolytes in aqueous organic redox flow batteries. Bigger connecting lines indicate more citations in the literature.

**Figure 8 molecules-27-00560-f008:**
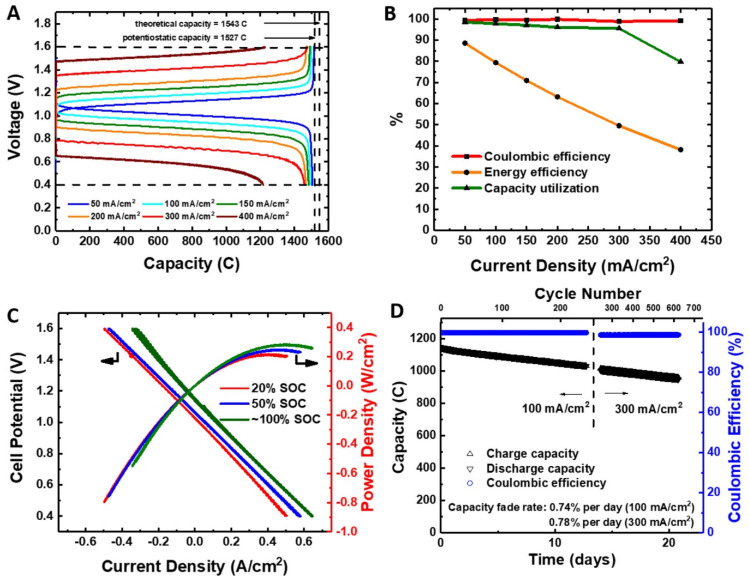
Full cell characterization of 0.5 M bislawsone negolyte in 1 M KOH and 0.3 M potassium ferrocyanide and 0.1 M potassium ferricyanide posilyte in 1 M KOH solution. (**A**) Galvanostatic charge–discharge voltage profiles from 50 to 400 mA/cm^2^. (**B**) Efficiencies and capacity utilization versus current density. (**C**) Cell load curves at 20%, 50%, and 100% SOC. (**D**) Coulombic efficiency and charge and discharge capacities during mid-term galvanostatic cycling with potentiostatic hold. Reprinted with permission from ACS Energy Lett. 2019, 4, 8, 1880–1887. Copyright © 2022, American Chemical Society.

**Figure 9 molecules-27-00560-f009:**
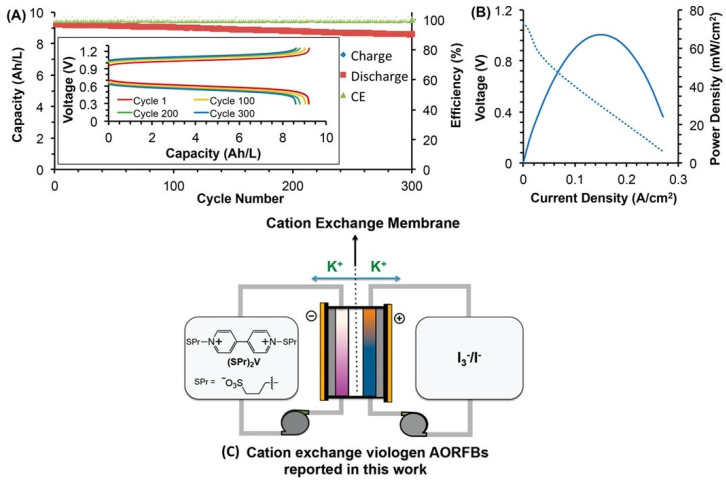
(**A**) Extended cycling data of the (SPr)_2_V/KI AORFB showing charge-discharge capacity and coulombic efficiency during galvanostatic cycling at 60 mA/cm^2^. Inset: Representative charge and discharge curve from the experiment. (**B**) Polarization (primary axis) and power density data (secondary axis) collected at 100% SOC. Conditions: negolyte, 0.5 M (SPr)_2_V in 2.0 M KCl; posilyte, 2.0 M KI in 2.0 M KCl, Nafion 212 cation exchange membrane. (**C**) Battery design of cation exchange viologen AORFBs. Adopted with permission from CS Energy Lett. 2018, 3, 3, 663–668 [117]. Copyright © 2022, A3merican Chemical Society.

**Figure 10 molecules-27-00560-f010:**
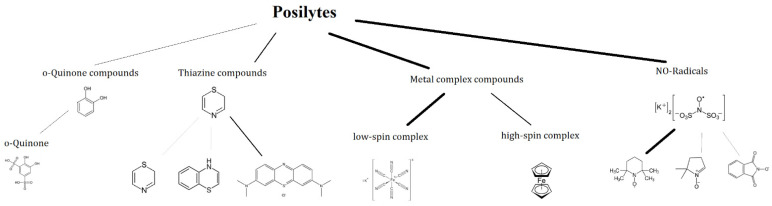
Family tree of the chemical motives for organic redox-active molecules, which could act as posilytes in aqueous organic redox flow batteries. Bigger connecting lines indicate more citations in the literature.

**Figure 11 molecules-27-00560-f011:**
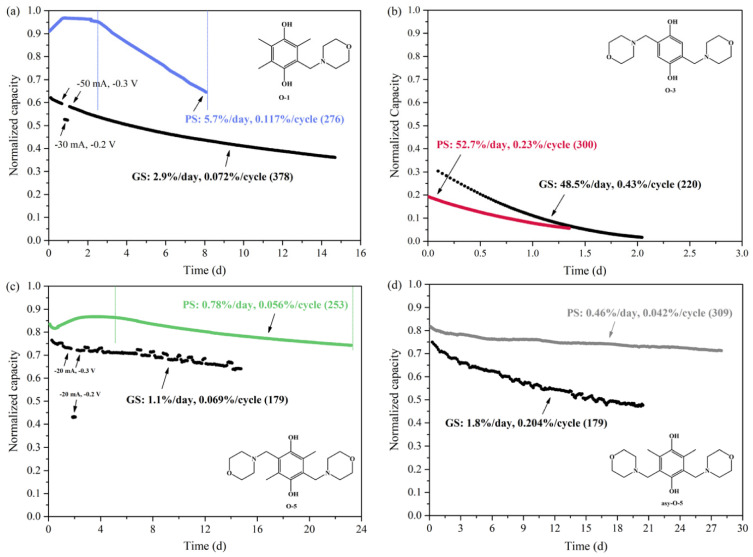
(**a**–**d**) Normalized discharge capacity of the symmetric cell tests of various modified benzoquinones using potentiostatic (PS) and galvanostatic (GS) cycling methods. Adopted from Molecules 2021, 26(13), 3823 [143].

**Table 1 molecules-27-00560-t001:** Important parameters of electrolytes and how they relate to battery key performance indicators: (a) energy density, (b) power density, (c) lifetime, (d) environmental impact, (e) efficiency, and (f) price.

Physicochemical, Chemical and Biological Properties of Electrolytes					
1	Solubility	a (b, f)	6	Dynamic viscosity	e (b)
2	Redox potential	a (b, f)	7	Permeation	a (c)
3	Redox kinetic	b (e)	8	Effect of temperature	c, e
4	Chemical/electrochemical stability	c	9	Aquatic and human toxicity	d
5	Ionic conductivity	b (e)	10	Abundance of materials	f

**Table 2 molecules-27-00560-t002:** Physico-chemical properties [8,9,10,11,12] of typical organic solvents in electrochemistry arranged according to their permeability (ε).

	ε at 20 °C	η at 20 °C [cP]	DN [kcal/mol]	AN [kcal/mol]	π*	ρ at 20 °C [g/cm^3^]
Formamide (FA)	109	3.75	24	39.8	97	1.13
**Water**	**80.1**	**1.002**	**18**	**54.8**	**109**	**0.998**
Propylenecarbonate (PC)	64.9	2.53	15.1	18.3	83	1.21
Dimethylsulfoxide (DMSO)	46.5	1.99	29.8	19.3	100	1.10
g-Butyrolactone (GBL)	39.1	1.73	18	17.3	87	1.13
Dimethylformamide (DMF)	36.7	0.80	26.6	16.0	88	0.94
N, N′-Dimethylpropyleneurea (DMPU)	36.1	2.93	15	42.0	-	1.06
Acetonitrile (AN)	35.9	0.34	14.1	18.9	75	0.78
Hexamethylphosphoramide (HMPA)	30.0	3.53	38.8	10.6	-	1.02
1,3-Dioxalane (DO)	7.3	0.60	-	-	69	1.06
Dimethoxyethane (DMO)	7.2	0.46	23.9	10.2	53	1.38
Methylacetate (MA)	6.7	0.36	16.5	10.7	60	0.94
Ethylacetate (EA)	6.0	0.42	17.1	9.3	55	0.90

**Table 3 molecules-27-00560-t003:** Thirty identified critical raw materials from the list by the EU Commission—highlighted are materials used in different lithium batteries (red) and flow batteries (green).

Critical Raw Materials									
01	Antimony	07	**Cobalt**	13	HREEs	19	Natural rubber	25	**Silicon metal**
02	Baryte	08	Coking coal	14	Indium	20	Niobium	26	Tantalum
03	**Bau** **xite**	09	**Fluor** **spar**	15	**Lithium**	21	PMGs	27	Titanium
04	Beryllium	10	Gallium	16	LREEs	22	**Phospha** **te rock**	28	**Vanadium**
05	Bismuth	11	Germanium	17	Magnesium	23	**Phosp** **horus**	29	Tungsten
06	Borates	12	Hafnium	18	**Natural** **graphite**	24	Scandium	30	Strontium

## Data Availability

No new data were created or analyzed in this study. Data sharing is not applicable to this article.

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
