# Peer review of "Family Tree for Aqueous Organic Redox Couples for Redox Flow Battery Electrolytes: A Conceptual Review"

_molecules, 2022, doi:10.3390/molecules27020560_

Round 1
Reviewer 1 Report
In this manuscript, the up-to-date overview of the redox organic compounds groups tested for application in aqueous RFBs are summarized, since RFBs have been shifted from metal-ion based electrolytes to soluble organic redox-active compounds. This review may provide an introduction to the field of organic RFBs and an overview of current achievements.
The content of this manuscript meets the reading interests of the readers of the Molecules. Therefore, I suggest a major revision to further improve the quality of the current version of the manuscript.
- Keywords, ‘aqueous’, ‘energy storage’ and ‘electrolyte’ should also be added in order to further attract a broader readership.
- Line 31, when referring to renewable energy, the disadvantage should also be described briefly, that is why an energy storage system is needed. ‘For example, by storing the intermittent renewable energy into powerful and reliable energy storage devices, the renewable energy power supply with a longer service life can be developed without putting pressure on the earth's resources’ [Journal of Power Sources 493 (2021): 229445].
- Line 57, ‘With current raw material prices, a VRFB cannot be built below this price. Organic electrolytes could change the costs structure...’The relationship between the front and back sentences here turns too fast and is not very smooth logical. For example, since the price of vanadium is so high, why not try the aqueous RFBs of other elements? Why do we have to go directly to the organic RFBs? I consider cost-effective Zinc and Iron-based flow batteries should be introduced briefly [Advanced Materials 31.50 (2019): 1902025; Journal of Power Sources 493 (2021): 229445], and their limitations as well before jumping to organic electrolytes.
- Line 70, ‘The biggest problem of aqueous organic electrolytes so far is the comparably low energy density, as mixed electrolytes of aqueous organic redox couples hardly exceed 13 Wh/l (≈ 1mol/l e–).’Here, some energy density values of VRFB or other aqueous RFBs should be provided for further comparison. Otherwise, it is difficult for readers to have an intuitive feeling and comparison through only one number. In the latter part, (≈ 20-40 Wh/l) belongs to which type of RFB should be clear, and this should be energy density (not capacity rate, see Line 75).
- Line 108, ‘During the conversion the charges exchanged between the two reaction surfaces (electrodes) lead to changes in the redox state of the storage material.’
Since electrode materials are mentioned and the redox reaction required to realize battery charge-discharge occurs here, it is necessary to briefly introduce the properties and challenges of ideal electrode materials. For example, the commonly used are carbon fibre-based electrodes (cloth, paper, or felt), and they are intrinsic hydrophobic. So pretreatment is generally required, including thermal and acid treatment, to improve its hydrophilicity [International Journal of Energy Research 44.5 (2020): 3839-3853]. At the same time, in order to improve the reaction activity and kinetics of the electrode, the electrode is usually modified appropriately, such as introducing appropriate inorganic particles [Electrochimica Acta 268 (2018): 59-65; Electrochimica Acta 336 (2020): 135646].
- Line 428, ‘In most electrolytes mid-concentrated strong acids or bases are used as electrolyte, as they offer the highest conductivity of all ionic substances. ’What is the exact concentration ranges of the mid-concentrated should be provided. Is it the latter mentioned ‘4-5 mol/l’? And why higher concentration is not selected, since that may provide the highest conductivity? The description ‘At higher concentrations, more water molecules are bound to the counterion, restricting the free movement of water molecules’, is it a common rule? If so, why in many VRFB systems 3M-4M H2SO4 are adopted, which is higher than 4-5 mol/L protons?
- Line 504, ‘In RFB, mostly homogeneous ion-exchange polymeric membranes are used providing a good compromise between high ionic conductivity and low permeability for electroactive species.’This sentence does not explain very accurately. What is the homogeneous membrane? I consider Nafion is also a homogeneous membrane, but it does not provide a good compromise between conductivity and permeability. Normally, composite membranes are used for RFBs with good selectivity (ratio between conductivity and active species permeability), for example, by polymer blending method [Journal of Membrane Science 601 (2020): 117906].
- Line 516, ‘The significantly increased size of the polymers enables to use of cheap size-exclusion porous membranes, which can reduce the system investment costs.’What is the increased size of the polymers? I consider it should be the increased size of the pores in the polymers since porous membranes are used and the pores appear that distinguish porous membranes from the traditional dense ion-exchange membranes.
- The summary of organic active substances in aqueous RFB systems by the family tree of Section 3 is very comprehensive and in place. However, this part is mainly a language description. Figures mostly represent the chemical structure and composition of active substances.
I consider in this part, the performance of RFB single-cell is very important and most interesting to readers. Therefore, I suggest adding one or two charge-discharge curves of active substances with the best performance/multi cycles capacity attenuation, CE and EE diagrams, as well as a table to summarize and compare the battery performance of various active substances.
Author Response
Thank you for your useful comments and suggestions on our manuscript. In the attachment, there are presenting replies and comments to the reviewer’s remarks.

Reviewer 2 Report
The manuscript presents an up-to-date overview of the redox organic compounds groups tested for application in aqueous RFB. The importance of supporting electrolytes selection, the limits for the aqueous systems and potential synthetic strategies for redox molecules are highlighted. The different organic redox couples described in the literature are grouped in a ‘family tree’ for organic redox couples.
I consider the content of this manuscript will definitely meet the reading interests of the readers of the Molecules journal. However, the discussion and explanation should be further improved. And the relevant results are all written descriptions, and there are almost no figures in the full text. It is monotonous and boring for readers as a review paper.
Therefore, I suggest giving a minor revision and the authors need to clarify some issues or supply some more experimental data to enrich the content. This could be a comprehensive and meaningful work after revision, especially paying attention to the grammar errors as well.
My detailed comments can be found in a separate PDF file.

Author Response

(The authors gave the same response as above.)

Round 2
Reviewer 1 Report
I have carefully read through the updated version of the manuscript. And I can conclude that the authors have addressed all of my previous comments. I consider this manuscript can be published in its current form.